# MARGE: Improving Math Reasoning with Guided Exploration

**Jingyue Gao** [1]  **Runji Lin** [2]  **Keming Lu** [2]  **Bowen Yu** [2]  **Junyang Lin** [2]  **Jianyu Chen** [1 3]

## Abstract

Large Language Models (LLMs) exhibit strong potential in mathematical reasoning, yet their effectiveness is often limited by a shortage of high-quality queries. This limitation necessitates scaling up computational responses through self-generated data, yet current methods struggle due to spurious correlated data caused by ineffective exploration across all reasoning stages. To address such challenge, we introduce **MARGE**: Improving **Ma**th **R**easoning with **G**uided **E**xploration, a novel method to address this issue and enhance mathematical reasoning through hit-guided exploration. MARGE systematically explores intermediate reasoning states derived from self-generated solutions, enabling adequate exploration and improved credit assignment throughout the reasoning process. Through extensive experiments across multiple backbone models and benchmarks, we demonstrate that MARGE significantly improves reasoning capabilities without requiring external annotations or training additional value models. Notably, MARGE improves both single-shot accuracy and exploration diversity, mitigating a common trade-off in alignment methods. These results demonstrate MARGE's effectiveness in enhancing mathematical reasoning capabilities and unlocking the potential of scaling self-generated training data.

## 1. Introduction

Large Language Models (LLMs) have demonstrated remarkable capabilities in text generation and instruction following through post-training techniques such as Reinforcement Learning (RL) (Stiennon et al., 2022; Ouyang et al., 2022).

[1]Institute for Interdisciplinary Information Sciences, Tsinghua University, Beijing, China [2]Alibaba Group [3]Shanghai Qi Zhi Institute, Shanghai, China. Correspondence to: Runji Lin <linrunji.lrj@alibaba-inc.com>, Jianyu Chen <jianyuchen@tsinghua.edu.cn>.

*Proceedings of the 42nd International Conference on Machine Learning*, Vancouver, Canada. PMLR 267, 2025. Copyright 2025 by the author(s).

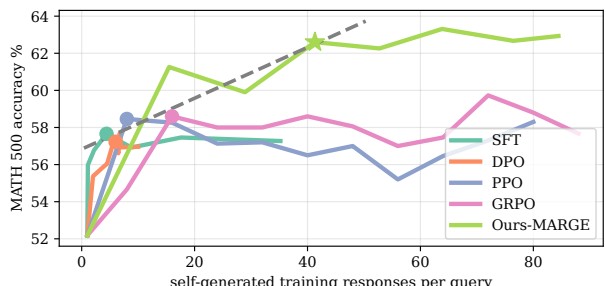

*Figure 1.* The y-axis represents the accuracy on MATH500, and the x-axis represents the number of self-generated responses for training Qwen2-7B-Instruct. The dots and the star show when models start to converge, and the dashed line exhibits their scaling trends as the generation amount for training increases. The top-right zone is preferred, as we can easily scale generation and achieve better performance when queries are limited. By improving the exploration process, MARGE enables the scaling of self-generated responses for training and improves reasoning ability. We discuss more about the scaling trend in Appx. D.

Despite these advances, improving LLM performance in long-horizon reasoning tasks, particularly mathematical problem solving, remains challenging due to two fundamental challenges: the scarcity of high-quality training data and the inherent difficulty of exploring the vast space of possible reasoning paths.

While Self-training approaches (Zelikman et al., 2022) partially address data scarcity by self-generating abundant training examples, they struggle to scale computation during training to improve reasoning. This limitation is evident in our observations: model performance initially improves during self-training but quickly plateaus or even declines as training data increases. These trends, consistent with Setlur et al. (2024), indicate spurious correlations between reasoning steps in self-generated datasets.

As explained below, we argue that insufficient exploration causes spurious correlated datasets. Our primary research question now becomes: **How can we enhance exploration across all reasoning stages to generate higher-quality training data, improve credit assignments, and enable scaling of self-generated training pipelines?**

Fig. 3 exhibits a challenge in mathematical reasoning and helps explain why insufficient exploration causes spurious correlations: even when a model correctly executes all pre-

vious steps, it still might fail the final steps on average. This highlights the need for reasoning data that accurately assigns credits to later stages without introducing spurious correlations. As the number of reasoning steps increases, the whole search space grows exponentially. This growing search space makes it exponentially complex to find this kind of data as the steps get progressively further back (e.g., for DPO, we will need data that only diverges at the particular step but remains the same at previous steps).

To address these limitations, we introduce **MARGE** (**Ma**th **R**easoning with **G**uided **E**xploration), a framework that systematically improves mathematical reasoning through guided exploration. MARGE leverages existing solutions as guidance, completing intermediate reasoning steps to generate diverse, high-quality training datasets. This approach decomposes the exponentially complex exploration problem into manageable sub-problems while efficiently utilizing computational resources. By constructing datasets with common solution prefixes and strategically selecting guidance (correct solutions for complex problems, incorrect ones for simpler cases), MARGE achieves comprehensive coverage across reasoning stages and improves credit assignment–particularly for later steps where traditional approaches often fail.

Our investigation reveals two fundamental insights about guided exploration in mathematical reasoning. First, iteratively running MARGE demonstrates that strategic guidance improves exploration during self-generation. It enables scalable improvements across diverse tasks, surpassing the performance plateaus encountered by prior methods. Second, and more interestingly, unlike traditional post-training methods that trade-off generation diversity (pass@$k$) for single-shot accuracy (pass@1) (Wang et al., 2024a), MARGE mitigates the issue and even might enhance both abilities simultaneously. This improvement arises from MARGE's systematic coverage of diverse reasoning strategies across all stages, effectively expanding the model's solution repertoire while reinforcing successful reasoning patterns.

Several recent works that try to enhance the reasoning abilities of LLMs, though not explicitly focused on exploration across reasoning stages, share our insights into enforcing exploration at different stages. Setlur et al. (2024); Xie et al. (2024); Lai et al. (2024); Lu et al. (2024b) explored step-level supervision, attempting to identify and correct erroneous steps in reasoning. However, these approaches usually depend on external supervision or heuristics, which may fail to capture the diversity at all reasoning stages compared to MARGE, limiting their scalability. Monte Carlo Tree Search (MCTS)-based methods (Chen et al., 2024; Zhang et al., 2024; Feng et al., 2023; Xie et al., 2024) generate high-quality datasets by rigorously searching and exploring reasoning strategies. While they focus on finding optimal

solutions, MARGE explicitly aims to understand both successful and unsuccessful reasoning patterns and provide a better exploration strategy for LLM reasoning. Additionally, MARGE operates efficiently by only inferring from the base LLM, eliminating the need for additional value models or searching computation required by MCTS.

We conduct extensive experiments across various base models to demonstrate the effectiveness of MARGE. When applied to Qwen2-7B-Instruct, it achieves significant performance gains across benchmarks like MATH (+7.90%), GSM8k (+3.03%), CollegeMath (+13.64%), and OlympiadBench (+5.23%), using pure chain-of-thought reasoning. Furthermore, MARGE improves intrinsic exploration, as evidenced by the higher performance gaps on pass@64.

In summary, our contributions are as follows:

- We introduce MARGE, a novel framework for guided exploration in LLM mathematical reasoning, laying the foundation for scaling the self-training pipeline;

- Leveraging solution-guided exploration, we make it possible to find more high-quality data, resulting in better exploration and credit assignment for complex multi-step mathematic problems;

- We demonstrate MARGE's efficacy in improving reasoning accuracy and generation diversity across multiple base models and benchmarks.

## 2. Related Works

### 2.1. LLM for Mathematical Reasoning

Previous works have proposed various methods to enhance the long-horizon reasoning capabilities of LLMs, particularly in mathematical problems. One line of research(Wei et al., 2023; Chen et al., 2023) focuses on eliciting the reasoning potential of LLMs with carefully designed prompts without updating the model. On the other hand, some approaches involve training LLMs with additional data. Lu et al. (2024a); Gou et al. (2024); Yu et al. (2024); Liu et al. (2024a) synthesize new problem-solution pairs to fine-tune a base model. Furthermore, employing process-level supervision during RL or Best-of-N sampling has been explored (Wang et al., 2023; Jiao et al., 2024; Lightman et al., 2023; Wang et al., 2024b; Luo et al., 2024), which, while effective, often necessitates extensive generation and verification efforts. Our approach leverages only the policy model to generate datasets, thus simplifying the acquisition process.

Some recent reasoning models like o1 (Jaech et al., 2024), QwQ-32B-preview (Qwen, 2024), and deepseek-r1 (DeepSeek, 2024) greatly improve LLM's reasoning capacity through test-time scaling. Some other methods (Yao et al., 2023; Qi et al., 2024; Xiang et al., 2025) also discuss

how to achieve such test-time scaling through system-2 planning or carefully designed reasoning pipelines, which accomplish similar improvements with smaller general models. Our method is perpendicular to these works as we address exploration issues and help scale self-generated responses during training. We also believe that they can be combined with our method in the future to enhance LLM's reasoning ability further.

## 2.2. Reinforcement Learning for LLM Reasoning

Reinforcement Learning (RL) has effectively been applied to align LLMs with human preferences across various tasks, including instruction-following and summarization (Ouyang et al., 2022; Stiennon et al., 2022). Recent works (Yang et al., 2024a;b; Jiao et al., 2024) apply these methods to mathematical reasoning tasks. Xi et al. (2024) introduces reverse curriculum learning into math reasoning. Setlur et al. (2024); Lai et al. (2024); Lu et al. (2024b) generate pairwise preference datasets from intermediate states and employ DPO to train the model. Chen et al. (2024); Feng et al. (2023); Xie et al. (2024); Zhang et al. (2024) either learn a step-level value function or directly fine-tune LLM through sampling with Monte Carlo Tree Search. Our method proposes a more efficient and natural exploration process for LLM reasoning, yielding a simple yet effective training pipeline that does not need external supervision or extra models.

## 3. Methods

This section introduces our method. We first present our problem formulation in Sec. 3.1, followed by the three main components of our proposed method in Sec. 3.2, Sec. 3.3, and Sec. 3.4. The general pipeline of our method is illustrated in Fig. 2 and Algo. 1. We enable the policy to cover the solution space better, persistently improving its reasoning ability as the generated responses scale when running MARGE iteratively. We provide additional theoretical analysis on how MARGE works in Appendix C, examples in Appendix F, and failure analysis in Appendix G.

### 3.1. Problem formulation

**Reinforcement Learning for LLMs** Reinforcement Learning (RL) in the field of LLMs mainly involves two stages: training a reward model and optimizing this model using RL algorithms such as PPO (Schulman et al., 2017) and REINFORCE (Sutton et al., 1999). An alternative approach, Direct Preference Optimization (DPO) (Rafailov et al., 2024), simplifies this process by learning directly from a preference dataset. Given a dataset that consists of a chosen and a rejected response for all prompts, DPO directly updates the policy $\pi$ with a loss analogous to RLHF.

**State-level mathematical reasoning** Let $a \oplus b$ represent the concatenation of strings $a$ and $b$. We conceptualize the reasoning process as a Markov Decision Process (MDP) $\langle S, A, P, r \rangle$, where states $s \in S$ and actions $a \in A$ sampled from policy $\pi_\theta$, and transition dynamics formulated as $s_{t+1} = s_t \oplus a_t$. Each response $y$ can be decomposed into multiple reasoning steps $y = m_1 \oplus \cdots \oplus m_n$, delineated by specific delimiters or simply splitting the response. An intermediate reasoning state $s_i$, also a prefix of $y$, can then be formulated as $s_i = m_1 \oplus \cdots \oplus m_i$, including the first state $s_0$ which is an empty string. The terminal state $s_n = y$ encompasses the final answer and, therefore not considered an intermediate state. The reward function $r(x, y) = 1$ if the full response $y$ correctly answers the question $x$. Correspondingly, the value of state $s_i$, which equals the Q value of $(s_{i-1}, a_{i-1})$, can be computed as

$$Q^\pi(s_{i-1}, a_{i-1}) = V^\pi(s_i) = \mathbb{E}_{\tau \sim \pi(x \oplus s_i)} r(x, s_i \oplus \tau). \quad (1)$$

### 3.2. Output Reward MC as Value Estimation

To estimate the value of an intermediate state $s$, we use the Monte Carlo (MC) simulation to calculate $V^\pi(s_i)$ in Eq. 1. Starting from state $s_i$, we generate $n$ completions from the current policy $\pi$, which we denote as $\tau = m_{i+1} \oplus \cdots \oplus m_n \sim \pi(x \oplus s_i)$. The correctness of these completions is assessed to estimate the true value $V^\pi$ as

$$\hat{V}^\pi(s_i) = \frac{1}{n} \sum_{j=1}^n r(x, s_i \oplus \tau_j). \quad (2)$$

This MC estimation method has several advantages over training a separate value model, such as a Process Reward Model. First, employing MC simulation avoids the need for additional model training and simplifies implementation(Wang et al., 2023; Jiao et al., 2024). Besides, MC estimation provides an on-policy estimation, allowing responses to be used in later policy improvement.

### 3.3. Hit-guided Exploration

In this subsection, we discuss how utilizing the intermediate states of a self-generated hit can improve the exploration strategy of LLMs. Previous works (Salimans & Chen, 2018; Florensa et al., 2017; Xi et al., 2024) on RL and LLM reasoning show that starting from a demonstration state that is close to the terminal can reduce the difficulty of acquiring reward signals. Inspired by these findings, we propose to sample responses by continuing from all intermediate states of a selected response, which we call hit-guided exploration. This enables better exploration over the entire reasoning horizon.

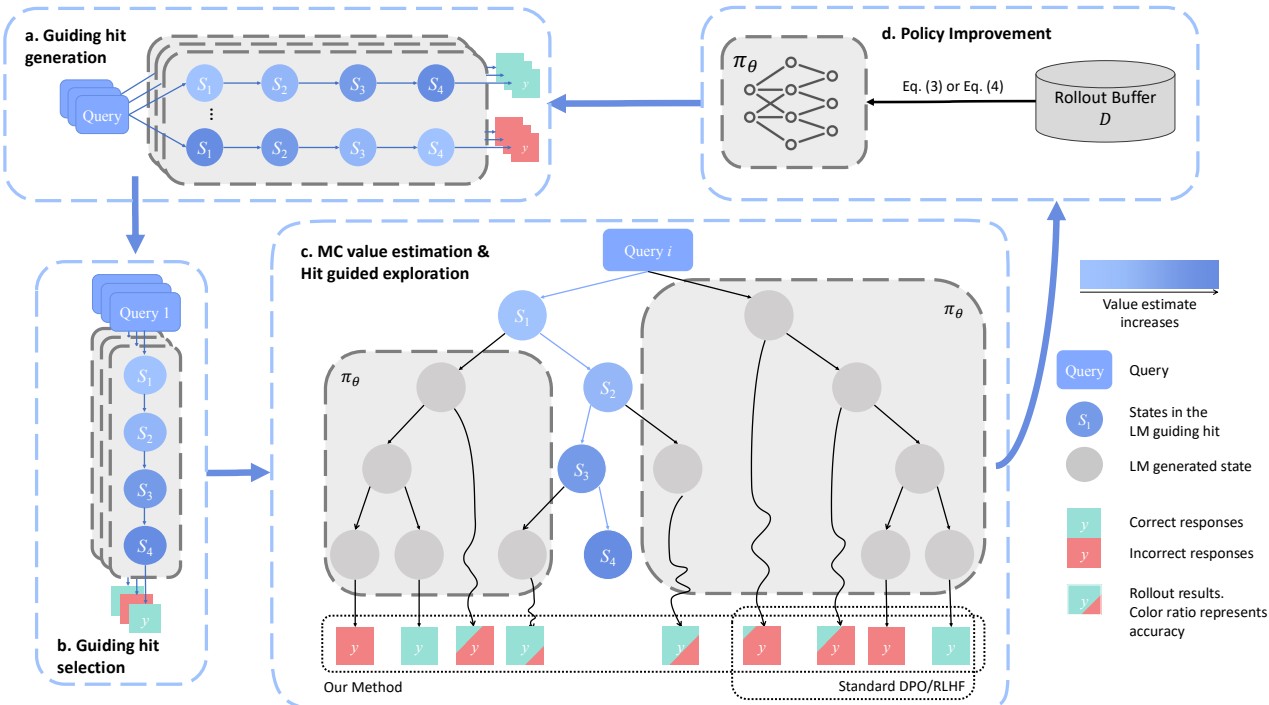

*Figure 2.* Overview of our method MARGE, which includes four stages. **(a)**: Generate multiple responses from the current policy $\pi^{(i)}$ as candidates for guidance and judge their correctness. Starting from the second iteration, we can directly leverage sampled responses from stage (c). **(b)**: Among all candidate solutions, we select one for each query as guidance according to Sec. 3.3. **(c)**: Perform a continuation of all states in the guide solution to complete the exploration (Sec.3.3) and value estimation (Sec.3.2). The collected data is utilized in stage **(d)** for training and stage **(a)** in the next iteration as well. **(d)**: Having fully explored the state space in **(c)**, we first form the rollout buffer, then optimize the current policy, and finally acquire the policy $\pi^{(i+1)}$ for the next iteration as described in Sec. 3.4.

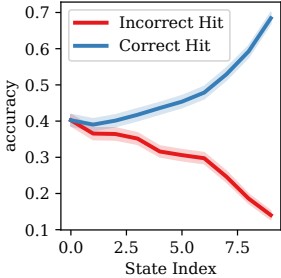

*Figure 3.* Average accuracies when starting from different intermediate states of correct solutions (blue) and incorrect ones (red) with Qwen2-7B-Instruct. A larger state index indicates being closer to the end. On average, completing from a correct (incorrect) state increases the portion of correct (incorrect) answers, which boosts the exploration of more training data.

**Hit-Guided Exploration** We assume a guide solution is available for each question, which can be acquired by sampling from the policy. Continuing from the solution's intermediate reasoning states, our explored responses naturally share common prefixes. This pattern implicitly helps assign credits to generated parts since the first steps are shared between completions. The key differences in correctness can, therefore, be identified in the later steps. In the meantime, as the first few steps are fixed, the search space

is greatly reduced, making exploration much easier.

Furthermore, hit-guided exploration can potentially increase the amount of valid training data. As illustrated in Fig. 3, the precision of completing an intermediate state increases on average as the state gets closer to the end of a correct response. This result, also proven in Prop. C.1, indicates that hit-guided exploration with a correct response increases the ratio of correct responses among all generations. The opposite also holds for incorrect ones. This shows that we can improve the number of desired responses if appropriate solutions are leveraged as guidance.

**Hit Selection and Update** When aligning LLMs, in particular reasoning, datasets that contain both positive and negative responses are preferred. Such datasets provide rich contrastive information regarding possible reasoning failure and corrections, thereby greatly improving data efficiency (Setlur et al., 2024). Therefore, the guiding hit must be carefully selected to find more paired responses. To accomplish this, we design a simple yet effective heuristic for selecting the guiding solution. We refer to queries with an estimated value greater than $0.5$ as easy queries and the others as hard queries. We then randomly select a correct solution for each hard query and an incorrect one for each

easy question. This solution is based on the simple intuition of increasing the likelihood of finding the right answer to a difficult question and possible failure cases to an easy one. Although this is a rather simple strategy, we demonstrate with experiments in Sec. 4.3.1 and theoretical analysis in Appendix C to show that, in most cases in reasoning, applying it helps increase the number of positive-negative pairs.

Another key factor is the on-policyness of generated data. Self-generated responses that are on-policy improve LLMs better as there is no distributional shift between responses and the optimized policy (Lai et al., 2024). Therefore, in each iteration from the first iteration, we update the guidance hit with the latest generation to ensure it is on-policy and that the explored data are equivalent to directly sampled from the latest policy.

### 3.4. Iterative Improvement

**Dataset Composition**   As the exploration process is done on all intermediate states of a response, all explored rollouts can be written as $D = \{(x_i, s_{ij}, y_{ijk}, r_{ijk})\}$, where $y_{ijk}$ means the $k$-th response generated from the $j$-th intermediate state $s_{ij}$ of query $x_i$, whose reward is $r_{ijk}$. We directly utilize $D$ with all rollouts for RL training. While for DPO training, one correct response $y^+$ and one incorrect response $y^-$ are sampled for each intermediate state to construct a preference dataset $D = \{(x_i, s_{ij}, y_{ij}^+, y_{ij}^-)\}$ for DPO. We filter states with estimated values that are either too high or too low to reduce noise and stabilize DPO training.

**Policy training**   Having collected the dataset, we update the current policy $\pi_\theta^{(i)}$ and use it as the reference with RL or DPO. For DPO, we use the correct response as $y_{\text{win}}$ and the incorrect one as $y_{\text{lose}}$ to calculate the DPO loss, which can be formulated as:

$$
\mathcal{L} = -\frac{1}{|D|} \sum_{(x, s, y^+, y^-) \in D} \big[ \log \sigma(\beta \log \frac{\pi_\theta(y^+|x \oplus s)}{\pi_{\text{ref}}(y^+|x \oplus s)} \\ -\beta \log \frac{\pi_\theta(y^-|x \oplus s)}{\pi_{\text{ref}}(y^-|x \oplus s)}) \big],
$$

$$(3)$$

where $\sigma$ is the sigmoid function, and $\beta$ is a parameter controlling the discrepancy of $\pi_\theta$.

For RL, we set the reward for the correct responses as 1 and 0 for incorrect ones. In addition to the vanilla REINFORCE loss, we add several modifications to improve training quality. We first replace the reward for the responses with their group relative advantages, which are calculated as in GRPO (Shao et al., 2024). We also add KL Divergence penalty to stabilize training, resulting in a loss function as

$$
\mathcal{L} = \frac{1}{|D|} \sum_{(x, s, y, r) \in D} [-\hat{r} \log \pi_\theta(y|x \oplus s)] + \beta \text{KL}(\pi_\theta || \pi_{\text{ref}}),
$$

$$(4)$$

where $\hat{r} = \frac{r - \text{mean}(r_i)}{\text{std}(r_i)}$ is calculated using $r_i$ of the responses gathered from the same intermediate state, and $\beta$ is a coefficient that balances policy gradient and the KL penalty.

## 4. Experiments

We design experiments to investigate three key research questions:

1. Does MARGE improve over baselines, and in which aspects, when controlling data and model parameters? (Sec. 4.2)

2. What are the benefits brought by introducing the hit-guided exploration strategy? (Sec. 4.3.1)

3. How do design choices affect the overall results of MARGE? (Sec. 4.3.2)

### 4.1. Experiment settings

**Models**   We utilize Qwen2-7B-Instruct, Qwen2.5-7B-Instruct (Yang et al., 2024a), LLaMa3.1-8B-Instruct (AI@Meta, 2024), and MetaMath-Mistral-7B (Jiang et al., 2023; Yu et al., 2023) as our backbone models to conduct experiments due to their widespread popularity and strong baseline performances. Additionally, we include Qwen2.5-Math-7B-Instruct (Yang et al., 2024b), a strong math-specific model.   They cover models of different types and math reasoning capacities, which we believe help demonstrate the effectiveness of our method.

**Baselines**   We first compare our method with algorithms applying vanilla exploration. We include SFT and DPO as our baselines on all models. Standard RL methods are also included to compare our method with other baselines. In particular, we compare our method with PPO (Schulman et al., 2017) and REINFORCE++ (Hu et al., 2024). We also include an updated version of REINFORCE that applies all the settings in our RL training, but with vanilla exploration. We describe it as GRPO in the following sections, as this objective is equivalent to GRPO (Shao et al., 2024) when the GRPO epoch is 1.

We also consider other potential exploration methods, including recent works incorporating step-level supervision or MCTS. We contrast our method with StepDPO (Lai et al., 2024) on Qwen2-7B-Instruct, which shares a similar idea of finding step-level supervision but through GPT-4 supervision. On MetaMath-Mistral-7B, we reproduce an MCTS-based exploration method, MCTS-DPO (Xie et al., 2024), as a baseline to showcase the efficacy of our exploration strategy over different types of related methods.

Besides controlled experiments above, we include several concurrent works (Cui et al., 2025; Liu et al., 2024b) that promote LLM reasoning through different perspectives, like

*Table 1.* Performance (pass@1 and pass@64 accuracy %) of different algorithms. The best result of each dataset is in **bold**, the second best one is underscored, and our model is marked in blue. We greedily sample the responses for each query and run them three times to report the average. Evaluation prompts are listed in Appendix E. Our method outperforms baseline methods on almost all datasets. More interestingly, our method improves pass@64 accuracy by a larger margin, indicating improvement of models' exploration abilities.

| Accuracy % | MATH | MATH500 | GSM8k | College Math | | OlympiadBench | |
|---|---|---|---|---|---|---|---|
| | pass@1 | pass@64 | pass@1 | pass@1 | pass@64 | pass@1 | pass@64 |
| Qwen2-7B-Instruct | 53.18 | 59.92 | 85.67 | 22.13 | 25.13 | 20.65 | 24.98 |
| SFT | 55.98 | 61.04 | 84.76 | 24.74 | 26.86 | 21.03 | 25.38 |
| DPO | 57.24 | 63.44 | 85.90 | 31.64 | 35.72 | 20.88 | 25.83 |
| PPO | 58.70 | 61.98 | 88.47 | 35.72 | 38.44 | 21.82 | 24.54 |
| REINFORCE++ | 59.81 | 63.58 | 88.19 | 35.58 | 38.28 | 24.49 | 25.62 |
| GRPO | 59.89 | 62.92 | 88.07 | 35.02 | 37.09 | 23.85 | 25.72 |
| StepDPO-HF | 57.78 | 63.54 | 87.90 | 30.92 | 32.36 | 22.91 | 24.19 |
| MARGE-DPO | 59.92 | 66.84 | 88.60 | 34.68 | 36.58 | 21.48 | 24.69 |
| MARGE-RL | **61.08** | **68.20** | **88.70** | **35.77** | **40.10** | **25.88** | **27.31** |
| MetaMath-Mistral | 28.68 | 34.66 | 75.28 | 17.56 | 21.67 | 7.10 | 12.09 |
| SFT | 28.60 | 38.28 | 75.94 | 17.31 | 21.72 | 6.67 | 14.07 |
| DPO | 26.70 | 38.68 | 74.50 | 14.84 | 20.94 | 5.43 | **15.10** |
| PPO | 27.78 | 32.54 | 78.11 | 17.81 | 20.90 | 7.06 | 10.56 |
| REINFORCE++ | 30.33 | 34.38 | 78.19 | 18.32 | 20.98 | 7.85 | 9.87 |
| GRPO | 30.76 | 37.08 | 79.27 | 18.38 | 22.39 | 6.67 | 11.55 |
| MCTS-DPO | 29.92 | 37.44 | 77.53 | 17.85 | 20.84 | 6.57 | 11.68 |
| MARGE-RL | **32.13** | **41.34** | **81.81** | **19.76** | **24.28** | **8.14** | 14.32 |
| Llama3.1-8B-Instruct | 49.96 | 70.33 | 85.97 | 28.11 | 37.34 | 16.34 | 34.47 |
| SFT | 50.72 | 64.96 | 86.37 | **30.03** | **39.10** | 16.89 | 34.41 |
| DPO | 50.36 | 71.54 | 86.68 | 27.39 | 36.77 | 15.75 | 36.29 |
| PPO | 50.50 | 65.18 | 85.06 | 26.38 | 34.22 | 15.75 | 28.39 |
| REINFORCE++ | 52.27 | 67.14 | 86.93 | 28.72 | 35.84 | **18.37** | 35.11 |
| GRPO | 51.22 | 71.00 | 86.58 | 28.04 | 37.82 | 15.41 | 34.37 |
| MARGE-RL | **54.23** | **72.36** | **88.36** | 28.94 | 38.19 | 17.33 | **38.61** |
| Qwen2.5-7B-Instruct | 75.30 | 79.62 | 91.89 | 40.41 | 44.48 | 36.00 | 41.77 |
| SFT | 75.17 | 80.33 | 92.27 | 41.09 | 44.39 | 38.12 | 41.23 |
| DPO | 75.03 | 76.97 | 92.06 | 40.57 | **44.78** | 38.23 | 42.76 |
| GRPO | 76.24 | 81.14 | 92.34 | 40.72 | 43.68 | 38.07 | 40.64 |
| MARGE-RL | **76.74** | **85.16** | **93.02** | **41.12** | 44.18 | **39.70** | **43.21** |

a revised multi-stage SFT process. MARGE enhances the underlying exploration process in RL to enable the scaling of self-generated responses for training. Therefore, they can potentially be combined, not as counterparts, for future enhancement in LLM reasoning.

**Datasets** For training, we start with the same subsets of MetaMathQA (Yu et al., 2024) and AQuA (Ling et al., 2017) as in StepDPO. Considering Qwen2.5-7B-Instruct and Qwen2.5-Math-7B-Instruct's already high performance in these tasks, we respectively randomly sample a subset of Omni-Math (Gao et al., 2024) and Big-Math(Albalak et al., 2025)'s training set.

To find a guide solution to each training query in our method, we generate 32 responses for each query in the training set at the beginning and select one as described in Sec. 3.3.

We filter the queries for which no suitable response was found, resulting in a curated dataset of approximately 8,500 questions paired with model-generated solutions. In the following rounds, the guidance solutions are selected similarly but directly from those generated in the previous round without additional sampling. If no appropriate solution exists for guidance, the guidance solutions for these queries are not updated.

For evaluation, we test our method on two widely adopted benchmarks: MATH (Hendrycks et al., 2021) and GSM8k (Cobbe et al., 2021), which include questions from grade school level to challenging competition problems. We also incorporate two more challenging datasets, Olympiad-Bench (He et al., 2024) and CollegeMath (Tang et al., 2024), to further test our model's generalizability on out-of-distribution challenging problems.

*Table 2.* Performance (average of 3 runs) of MARGE and concurrent works based on Qwen2.5-Math-7B models. The best result is in **bold**, and MARGE is marked in blue . MARGE significantly improves both the pass@1 accuracy and exploration ability over Qwen2.5-Math-7B-Instruct. It also showcases better or comparable performances to the most advanced works at 7B level.

| Accuracy % | MATH | MATH500 | | College MATH | | OlympiadBench | |
|---|---|---|---|---|---|---|---|
| | pass@1 | pass@1 | pass@64 | pass@1 | pass@64 | pass@1 | pass@64 |
| Qwen2.5-Math-7B-Instruct | 83.48 | 83.33 | 86.40 | 40.80 | 48.64 | 46.95 | 48.62 |
| PPO | 83.37 | 83.26 | 86.12 | 40.75 | 47.14 | 47.05 | 48.63 |
| MARGE-RL | **84.46** | **85.04** | **89.92** | 41.58 | 49.49 | 47.40 | 49.02 |
| ACEMath-7B (Liu et al., 2024b) | 83.13 | 83.42 | 85.72 | **42.76** | 50.32 | **48.68** | 50.45 |
| PRIME-EURUS-7B (Cui et al., 2025) | 80.08 | 80.70 | 88.58 | 40.99 | **58.96** | 48.22 | **51.97** |

**Implementation** Our experiments are done on 8 A100-80GB GPUs. When generating responses during training, we set the temperature at $0.8$ and the top p at $0.95$ to generate diverse responses, which are used to create training data sets or find self-generated solutions. Specifically, we use the vLLM (Kwon et al., 2023) engine to infer the policy model.

For DPO training and baselines, we set $\beta = 0.4$ and a global batch size of $256$. For RL training and baselines, we set the coefficient for KL loss as $0.01$ and sample $8$ responses per state (or query for RL baselines). For both our methods and baselines, we run for $10$ episodes and apply early-stopping. More hyperparameters and implementation details are in Appendix B.

### 4.2. Results

**Metrics** We evaluate models using two key metrics. First, we test single-shot Chain-of-Thought (CoT) accuracy (pass@1) across all datasets. Second, we assess multishot accuracy (pass@k) to measure a model's exploration ability. Pass@k represents the model's precision when given multiple attempts to solve questions, indicating its ability to explore diverse reasoning paths. We visualize the exploration ability by comparing pass@k improvement over pass@1 in Fig. 4. Due to computational constraints, we conduct pass@k testing on MATH500 (Lightman et al., 2023), which found effectively represent the complete MATH test set. We also evaluate pass@k on CollegeMath and OlympiadBench, excluding GSM8k due to its relative simplicity.

**Main Results** Our main results[1] in Tab. 1 and Tab. 2 demonstrate consistent performance gains across model architectures and benchmarks. MARGE also generalizes effectively to more difficult out-of-distribution test sets. It outperforms its vanilla, externally supervised, and MCTS-based exploration strategy counterparts by a large margin. As Tab. 1 shows, MARGE's improvements are relatively more significant on complex tasks like MATH, indicating

---

[1]For baseline methods, if an open-source model is available, we evaluate it as our model. Otherwise, we reproduce their method on our models with our training queries.

the necessity of adequately exploring challenging problems that require longer reasoning horizons.

Most notably in Tab. 1, Tab. 2 and Fig. 4, when MARGE improves both pass@1 and pass@k, the performance gap widens as $k$ increases, indicating enhanced exploration capabilities. This finding is further validated through continued training experiments. Running REINFORCE++ on the MARGE-trained Qwen2 model improves MATH accuracy to 62.34%, surpassing the 59.81% ceiling achieved when starting from the base Instruct model. This suggests MARGE training enhances both reasoning capabilities and solution diversity, allowing access to previously unexplorable response possibilities. This improvement suggests that during the MARGE training process, the model not only gains better reasoning capabilities but also develops additional diversity in its solution strategies, accessing previously unexplorable response types for the original model.

### 4.3. Ablation Study

To validate the effectiveness of MARGE, we conduct an extensive ablation study to justify key design decisions of our method. Ablation studies are conducted on the Qwen2-7B-Instruct model with the same training query set.

#### 4.3.1. BENEFITS OF HIT-GUIDED EXPLORATION

**It enables more training data** First, we show that our exploration method helps find more effective training data than vanilla exploration. We measure the number of valid pairs with the entropy of training data. Higher entropy indicates a larger ratio of possible correct-incorrect pairs among all data, which is beneficial to improve LLM as discussed in Sec. 3.3. We show how the entropy changes as the sampled responses grow in Fig. 5a. We compare vanilla exploration and hit-guided exploration with different hit selection strategies, including:

1. *Ours*: Selecting guidance with the heuristic discussed in Sec. 3.3, where we randomly select correct responses for hard questions and incorrect responses for easy ones.
2. *Random*: Randomly select a response.

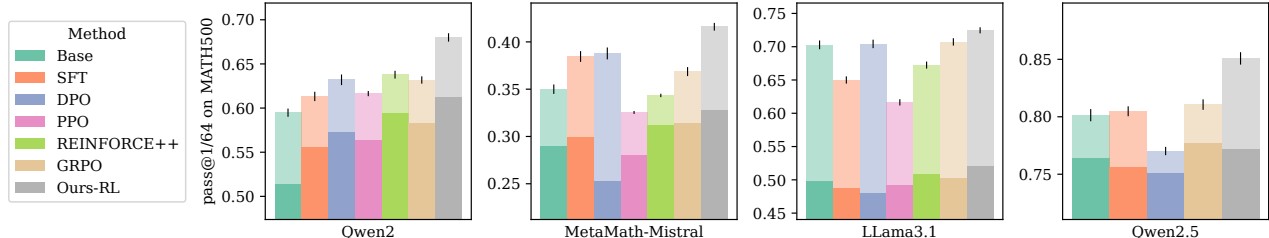

*Figure 4.* Pass@1 (solid) and pass@64 (shaded) of different methods on the MATH500 test set. Pass@64 indicates the ability to explore multiple reasoning paths. The figure displays the improvement pass@64 over pass@1 in the shaded area, symbolizing the models' ability to explore. When MARGE enhances both pass@1 and pass@k, the performance gap is larger as k grows. It demonstrates that, instead of trading off exploration abilities for pass@1 improvement, MARGE also enables a better exploration process than baselines and thus fundamentally improves exploration ability.

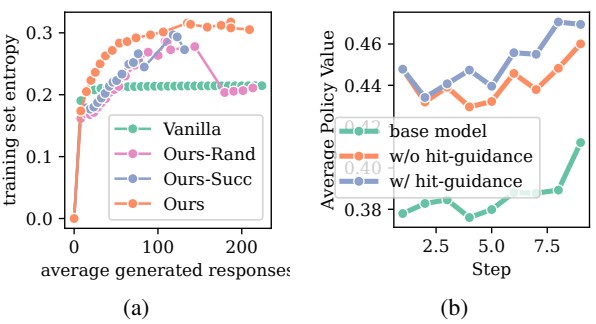

*Figure 5.* **(a)**: The change of training dataset entropy with number of responses sampled. Our method continues to find new useful pairs when generating more responses. As completing intermediate states does not yield full responses, we convert them based on the number of tokens. **(b)**: Hit-guided explored data improves average state value at every reasoning step compared to vanilla exploration, in particular later ones. The values are estimated over 32 Monte Carlo simulations.

3. *Succ*: Randomly select a correct response.

The results demonstrate that hit-guided exploration continues to uncover more pairs than vanilla exploration as more responses are generated. Besides, comparing different guidance selection strategies highlights the importance of selecting appropriate guidance for exploration. If not, the exploration performance might drop to the vanilla level.

**It helps improve on all reasoning steps** To further demonstrate the benefits of hit-guided exploration, we aim to validate that it improves over baselines at all reasoning steps, particularly the final ones. We test this hypothesis with two DPO experiments. The **only** difference is the way training sets are collected: one uses hit-guided exploration, while the other uses vanilla collection. We estimate the policy state value $V^\pi(s)$ of all intermediate steps from MATH500 collected by Qwen2-7B-Instruct. Aggregating over queries, we plot the estimated value of different steps in Fig. 5b and note that, while both versions of DPO significantly improve the state value on all steps, DPO with hit guidance data exhibit advantages starting from the third

reasoning step. When applying the same amount of data, our method covers all reasoning horizons and explores more effective data for optimizing LLM, particularly as the steps go backwards.

### 4.3.2. ABLATION STUDIES

**Generation Amount** Our method enables the discovery of more useful training data on the same set of queries, thus paving the way for successfully scaling self-generation for training. Therefore, to better control variables, we test how increasing the number of self-generated data for training influences the results of different algorithms. For SFT and DPO, we increase the number of responses (pairs) for each query; for RL baseline, we increase the number of rollouts per query. This is done so that the total generated tokens roughly match those in the MARGE training.

*Table 3.* Ablation study on the amount of training data. We include their best performance and the performance when they use the same amount of data as MARGE in Tab.1. We annotate the fraction of data used to achieve the best performance after `best:`. Results showcase the importance of exploration when scaling self-generated data.

|  |  | MATH | GSM8k | College Math | Olympiad Bench |
|---|---|---|---|---|---|
| SFT | best:1/4 | 57.66 | 86.65 | 24.94 | 22.22 |
|  | same data | 57.36 | 86.02 | 24.12 | 20.74 |
| DPO | best:1/16 | 57.24 | 85.90 | 31.64 | 20.88 |
|  | same data | 51.38 | 83.77 | 33.01 | 18.66 |
| REINFORCE | best:2/5 | 59.81 | 88.32 | 35.58 | 24.49 |
|  | same data | 59.79 | 87.94 | 33.43 | 24.15 |

The results in Tab. 3 demonstrate that adding more data to the baseline methods does not constantly improve the reasoning ability. In contrast, it might even deteriorate as spurious correlations occur. This further underscores the importance of efficient exploration and better credit assignment and the effectiveness of our method in achieving these goals.

*Table 4.* Ablation study on different guidance selection strategies. The results are single-shot accuracy (%). The results indicate that a better hit selection strategy achieves better exploration and improves the final result, and on-policy solutions play a crucial role in obtaining better exploration.

| Strategies | MATH | GSM8k | College Math | Olympiad Bench |
|---|---|---|---|---|
| Ours | 61.08 | 88.70 | 35.77 | 25.88 |
| Random | 60.21 | 88.6 | 35.06 | 24.96 |
| Succ | 59.91 | 87.59 | 35.06 | 23.51 |
| No Update | 59.54 | 88.09 | 35.54 | 24.36 |

**Hit Selection** As demonstrated above, correctly selecting the guidance trajectory helps improve exploration efficiency by finding more valid preference pairs. Here, we further validate its effects on final performances and compare different hit-selection strategies. We include another *No Update* off-policy baseline, where we fix the guidance solution once selected the same way as ours in the first round. We train MARGE with RL and calculate the accuracies of the four datasets as in Tab. 4. As we can infer from the table, our designed strategy performs better than other selection strategies. It can also be inferred that updating guidance helps improve performance, indicating the essentiality of on-policy solutions for exploration and policy improvement.

## 5. Computation Cost

Compared to vanilla exploration methods, MARGE naturally introduces more generations when other parameters are controlled (like number of samples per prompt, training set size, etc.). It changes the coefficient of time complexity, but not its asymptotic behaviour. Once the number of intermediate states $n$ is fixed, MARGE incorporates around $(n + 1)/2$ times more generation and training amount, as completing intermediate states generally requires fewer tokens. Statistics show it is approximately 3.3 on Qwen2, Llama3.1, and MetaMath, with around 5 states per query, and 4.9 on Qwen2.5, with around 8 states per query.

*Table 5.* The results of MARGE and some baselines are shown when MARGE uses less computation, such that the training time is roughly the same.

| | MATH | GSM8k | College Math | Olympiad Bench |
|---|---|---|---|---|
| PPO | 58.70 | 88.47 | 35.72 | 21.82 |
| REINFORCE | 59.81 | 88.32 | 35.58 | 24.49 |
| MARGE | 60.67 | 88.10 | 35.81 | 25.28 |

Tab. 5 presents the results of MARGE and some baselines on Qwen2 when MARGE uses less computation, such that the training time is roughly the same. MARGE exhibits certain advantages over baselines. However, as shown in Tab. 3, with more generations, baseline methods' performances saturate or even deteriorate, while MARGE's improved exploration ability allows it to continue improving.

While our method utilizes more generation computation, it is our goal and contribution to scale up the computation to make the most of the current query set. High-quality problems are getting harder to acquire. Therefore, we develop MARGE with a stronger exploration ability to find more high-quality training samples on the same query set. Possible ways to reduce the computation cost of MARGE exist, like removing unnecessary states from the Monte Carlo estimation.

## 6. Conclusion and Future Work

In this work, we identify and tackle the challenges of enhancing LLM reasoning through self-training. Our investigation identified ineffective exploration as a critical bottleneck in generating high-quality reasoning data, particularly for complex, multi-step tasks. To address this, we introduced MARGE, a novel method that systematically leverages self-generated solutions to improve data exploration and credit assignment across reasoning stages. Extensive experiments and ablations demonstrate that MARGE achieves substantial performance gains. Moreover, our method surpasses existing baselines by enhancing exploration diversity as well, exhibiting larger gains in pass@$k$ than pass@1. These results underscore the effectiveness of MARGE in improving exploration and scaling self-training pipelines for LLM reasoning. We discuss the explicit scaling effects of MARGE detailedly in Appendix D.

While MARGE demonstrates significant improvements, it still has several points for future improvements. First, MARGE's performance gains converge after adequate iterations (though more than baselines), which is likely due to the deterioration of generation quality during optimization. Addressing this issue without compromising reasoning gains is an interesting topic for future research. Second, though validated to be effective for mathematical reasoning, MARGE's exploration strategy is relatively simple and has not been tested in other domains. Extending it to diverse application tasks could further enhance its impact.

## Impact Statement

This paper presents work that aims to advance the fields of large language models (LLMs) and Reinforcement Learning (RL), in particular LLM reasoning. In particular, the developed algorithm makes it easier to explore the large state space and refine credit assignment during LLM reasoning. However, if our method is misused for inappropriate scenarios, it might cause LLMs to behave and respond unexpectedly.

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

# A. Algorithm

---
**Algorithm 1** MARGE
---
**Input:** Policy language model $\pi_\theta$; training query set $\mathcal{D}_\mathcal{P}$; number of episodes $M$; query batch size $B$; KL loss coefficient
    $\beta$; Monte Carlo simulation number $n$; initial responses generation number $n_1$; output reward function $r$.
  1: $\mathcal{D} \leftarrow$ generate_policy($\mathcal{D}_\mathcal{P}, \pi_\theta, n_1$)                  ▷ Generating $n_1$ hit candidates for all queries
  2: **for** $j = 1 \ldots M$ **do**
  3:      select_guidance_solution($\mathcal{D}$)            ▷ Sec 3.3: select a guiding solution for each question
  4:      **for** query batch $\mathcal{D}_i$ from $\mathcal{D}$ of size $B$ **do**
  5:          $\mathcal{S} \leftarrow$ get_states($\mathcal{D}_i$)               ▷ get states of the guidance solutions in $\mathcal{D}_i$
  6:          $\mathcal{A} \leftarrow$ generate_policy($\mathcal{S}, \pi_\theta, n$)   ▷ Sec 3.3: generate $n$ completions for all states in $\mathcal{S}$ with policy $\pi_\theta$
  7:          $V \leftarrow$ estimate_state_values($\mathcal{S}, \mathcal{A}$)             ▷ Sec 3.2: estimate state values
  8:          $\pi'_\theta \leftarrow$ train($\pi_\theta, D, V$)     ▷ Sec 3.4: train policy with objective Eq. 3 for DPO or Eq. 4 for RL
  9:          $\mathcal{D} \leftarrow$ update_hits($\mathcal{A}$)            ▷ Sec 3.3: update guidance candidates with latest responses
10:          $\pi_\theta \leftarrow \pi'_\theta$                            ▷ Use the updated policy
11:      **end for**
12: **end for**
**Output:** Trained policy $\pi_\theta$

---

# B. More Implementation details

### B.1. Our method

Our implementation is based on TRL (von Werra et al., 2020) and DeepSpeed (Rasley et al., 2020) framework. We utilize the vLLM (Kwon et al., 2023) engine to do inference.

We apply two ways to obtain the intermediate states of a guidance hit. The first way is to divide the response with special delimiters within the response the models generate, like `Step i:`. We leverage special prompts to generate such a pattern, as further discussed in Appendix E. In Qwen2-7B-Instruct, this way results in an average of $4.42$ states per question. In Qwen2.5-7B-Instruct, more difficult problems are incorporated for training, resulting in an average of $9.08$ states per question. Another way is to split the response directly based on the number of tokens. We evenly split 5 states from the guidance responses for other models. This is chosen based on the final performance of the models.

When collecting rollouts with hit-guided exploration, we set the following sampling parameters: temperature as $0.8$, top_p as $0.95$, top_k as $-1$. For each state, 8 responses are collected. During RL training, we set the learning rate as $1 \times 10^{-6}$ and batch size as 1024 to stabilize training. We set the coefficient for KL divergence as $0.01$ and train the model with a context length of 2048. We train on the collected dataset for 2 epochs within each iteration. During DPO training, we set $\beta$ to $0.4$. We set the learning rate as $5 \times 10^{-7}$ with a batch size of 256. Within each iteration, we train on the collected dataset for 4 epochs, with a maximum length of 2048.

### B.2. RL baselines

We train PPO and REINFORCE++ baselines with OpenRLHF (Hu et al., 2024). Some key parameters are listed in Tab. 6. Some parameters are different from our method as we tuned them for better performance. For example, the sampling parameters are different as PPO and REINFORCE++ with OpenRLHF fail to converge when the same ones are applied. We implement GRPO with TRL (von Werra et al., 2020) similar to our method, only differing in the way collecting rollouts. The parameters of GRPO are the same as our method.

### B.3. DPO

The training data for DPO are collected in the same way as in our method. We form preference pairs by randomly selecting correct and incorrect responses collected. We train DPO baseline with TRL (von Werra et al., 2020). We set $\beta = 0.4$ and a global batch size of 256. The learning rate is $5 \times 10^{-7}$.

*Table 6.* Some key parameters for RL baselines PPO and REINFORCE++.

|  | value |
| --- | --- |
| KL penalty coefficient | 0.01 |
| Samples per prompt | 8 |
| Actor learning rate | $5 \times 10^{-7}$ |
| Critic learning rate | $9 \times 10^{-6}$ |
| Discount Factor | 1 |
| Training batch size | 512 |
| Rollout batch size | 64(PPO)/128(REINFORCE) |
| Clip ratio | 0.2 |
| Maximum length | 2048 |
| Sampling temperature | 1.0 |
| Sampling top_p | 1.0 |

### B.4. SFT

The training data for SFT are collected in the same way as in our method. A randomly selected correct response for each query is used to run SFT. We set the learning rate to $5 \times 10^{-6}$, batch size to 128.

### B.5. Other baselines

**StepDPO (Lai et al., 2024)**   We directly utilize the open-source model provided[2].

**MCTS-DPO (Xie et al., 2024)**   We reproduce MCTS-DPO using its official repository[3] on the same training set as ours. We reproduce it on MetaMath-Mistral-7B as in their paper, with their default parameters. We did not include it in the Qwen models as we failed to run it successfully after many times of trials.

## C. Theoretical Analysis

In this section, we aim to provide a theoretical analysis of the foundation and effectiveness of MARGE.

In Proposition C.1, we first demonstrate that using a correct (or incorrect) solution as guidance increases the expectation of acquiring correct (or incorrect) responses, which lays the foundation for the motivation of hit-guided exploration. This intuitive phenomenon is empirically demonstrated in both previous works in RL (Florensa et al., 2017; Salimans & Chen, 2018) and in Figure 3. Here, we view such empirical conclusions from the viewpoint of theory.

**Proposition C.1.** *Suppose $S_1 \oplus \cdots \oplus S_n$ is a generated response. $R(q, S_1 \oplus \cdots \oplus S_n)$ is the reward function that gives 1 if and only if $S_1 \oplus \cdots \oplus S_n$ is a correct solution to q; otherwise $R = 0$. Under the condition that it is a randomly sampled correct response, we have:*

$$\mathbb{E}[R] \leq \mathbb{E}[R|S_1 \text{ is from a correct response}] \leq \cdots \leq \mathbb{E}[R|S_1, \ldots, S_n \text{ are from a correct response}].$$

*If it is an incorrect response, then:*

$$\mathbb{E}[R] \geq \mathbb{E}[R|S_1 \text{ is from an incorrect response}] \geq \cdots \geq \mathbb{E}[R|S_1, \ldots, S_n \text{ are from an incorrect response}].$$

*Proof.* Let's start with $\mathbb{E}[R] \leq \mathbb{E}[R|S_1]$ for correct responses. As in language generation, the process is of a discrete setting. What's more, as $S_1 \oplus \cdots \oplus S_n$ is randomly sampled from all correct responses, with a little bit of notation abusing for abbreviation, we can denote $\mathbb{E}[R|S_1$ is from a correct response] as $\mathbb{E}[R|S_0] = \sum_{S_1} P(R = 1|S_1)P(S_1|R = 1)$. Consider

---

[2]https://huggingface.co/xinlai/Qwen2-7B-Instruct-Step-DPO
[3]https://github.com/YuxiXie/MCTS-DPO

$\mathbb{E}[R|S_1] - \mathbb{E}[R]$, by Bayes' theorem, we have:

$$\begin{aligned}
&\mathbb{E}[R|S_1] - \mathbb{E}[R] \\
&= \sum_{S_1} P(R=1|S_1)P(S_1|R=1) - \sum_{S_1} P(R=1|S_1)P(S_1) \\
&= \sum_{S_1} P(R=1|S_1)\frac{P(R=1|S_1)P(S_1)}{P(R=1)} - \sum_{S_1} P(R=1|S_1)P(S_1) \\
&= \frac{\sum_{S_1} P(R=1|S_1)^2 P(S_1) - \left(\sum_{S_1} P(R=1|S_1)P(S_1)\right)^2}{P(R=1)} \\
&= \frac{\mathrm{Var}(P(R=1|S_1))}{P(R=1)} \geq 0,
\end{aligned} \tag{5}$$

so $\mathbb{E}[R|S_1] \geq \mathbb{E}[R]$. Similarly, we have $\mathbb{E}[R|S_1,\ldots,S_j] \geq \mathbb{E}[R|S_1,\ldots,S_{j-1}]$. Therefore, for a correct response,

$$\mathbb{E}[R] \leq \mathbb{E}[R|S_1] \leq \cdots \leq \mathbb{E}[R|S_1,\ldots,S_n]. \tag{6}$$

We denote a new reward function $R'$ for incorrect ones, such that $R' = 1 - R$ on all queries and responses. Therefore, similar to the deduction above, we have:

$$\mathbb{E}[R'|S_1] \geq \mathbb{E}[R'].$$

Subsituting back $R = 1 - R'$, we will have $\mathbb{E}[R|S_1] \leq \mathbb{E}[R]$. Iterating such a process, we will have

$$\mathbb{E}[R] \geq \mathbb{E}[R|S_1] \geq \cdots \geq \mathbb{E}[R|S_1,\ldots,S_n] \tag{7}$$

for incorrect guidance solutions. $\qquad\square$

In the following part, we aim to showcase the effectiveness in exploring more data of the hit-selection strategy in Section 3.3. First, under the hypothesis that the change of $\mathbb{E}[R|S_1,\ldots,S_i]$ in Proposition C.1 is linear to $i$, we show a sufficient condition for exploring more correct and incorrect pairs within the same number of responses in Proposition C.2. This linearity hypothesis is generally reasonable, shown by the empirical results in Figure 3 and Xi et al. (2024). The statistics in Figure 3 show general compliance with the conditions in Proposition C.2. The results in Figure 5a that directly reflect the dataset's pairs further validate the reasonableness of the hypothesis and the effectiveness of our method.

**Proposition C.2.** *Suppose $S_1 \oplus \cdots \oplus S_n$ is the generated response sampled for hit-guided exploration, and $k$ is an intermediate state. Suppose $\Delta = |\mathbb{E}[R|S_1,\ldots,S_j] - \mathbb{E}[R|S_1,\ldots,S_{j-1}]|$ is constant as $j$ changes. Let $\sharp_{\text{Hit-Guided}}$ be the number of valid correct-incorrect pairs introduced by hit-guided exploration, and $\sharp_{\text{Vanilla}}$ be the number of valid correct-incorrect pairs introduced by vanilla exploration. Then*

$$2k(k+1) \geq n(n+1), \quad k = \lfloor \frac{|1 - 2\mathbb{E}[R]|}{2\Delta} \rfloor,$$

*is a sufficient condition that hit-guided exploration introduces more valid pairs than vanilla exploration, i.e.*

$$\sharp_{\text{Hit-Guided}} \geq \sharp_{\text{Vanilla}}. \tag{8}$$

*Proof.* Suppose we sample $m$ responses in each state. As we only consider responses that share the same prefix, then the total number of valid pairs is

$$\sharp_{\text{Hit-Guided}} = \min(\mathbb{E}[R], 1 - \mathbb{E}[R])m + \sum_{i=1}^{n} \min(\mathbb{E}[R|S_1,\ldots,S_i], 1 - \mathbb{E}[R|S_1,\ldots,S_i])m.$$

For vanilla exploration, within $m(n+1)$ responses, the number of valid pairs is

$$\sharp_{\text{Vanilla}} = (n+1)\min(\mathbb{E}[R], 1 - \mathbb{E}[R])m.$$

Let's first consider the $\mathbb{E}[R] \leq \frac{1}{2}$ case, where $\sharp_{\text{Vanilla}} = (n+1)m\mathbb{E}[R]$.

When $S_1 \oplus \cdots \oplus S_n$ is an incorrect response, as in Proposition C.1, $\mathbb{E}[R|S_1, \ldots, S_i] \leq \mathbb{E}[R]$,

$$\sum_{i=1}^{n} \min(\mathbb{E}[R|S_1, \ldots, S_i], 1 - \mathbb{E}[R|S_1, \ldots, S_i])m = \sum_{i=1}^{n} \mathbb{E}[R|S_1, \ldots, S_i]m \leq n\mathbb{E}[R]m.$$

Therefore, using an incorrect response for hard queries with an estimated value smaller than $0.5$ cannot improve exploration.

On the other hand, let's consider the case when $S_1 \oplus \cdots \oplus S_n$ is a correct response. Since $\Delta$ is the slope for linear improvement in our hypothesis, by Proposition C.1, for arbitrary $i$, we have

$$p_i = \mathbb{E}[R|S_1, \ldots, S_i] = \mathbb{E}[R] + i\Delta. \tag{9}$$

In addition, we denote $p_0 = \mathbb{E}[R]$ for clearity. Consider $k = \lfloor \frac{|1-2\mathbb{E}[R]|}{2\Delta} \rfloor$. Then for all $i \leq k$, $\mathbb{E}[R|S_1, \ldots, S_i] \leq \frac{1}{2}$, and for all $i > k$, $\mathbb{E}[R|S_1, \ldots, S_i] \geq \frac{1}{2}$, and

$$\sharp_{\text{Hit-Guided}} = m \sum_{i=0}^{k} p_i + m \sum_{i=k}^{n} (1 - p_i).$$

To make $\sharp_{\text{Hit-Guided}} \geq \sharp_{\text{Vanilla}}$, we need to ensure that

$$\sharp_{\text{Hit-Guided}} - \sharp_{\text{Vanilla}} = m \sum_{i=0}^{k} p_i + m \sum_{i=k}^{n} (1 - p_i) - (n+1)mp_0 \geq 0. \tag{10}$$

Substitute Equation (9) into Equation (10), we have

$$\sharp_{\text{Hit-Guided}} - \sharp_{\text{Vanilla}} = m \sum_{i=1}^{k} i\Delta + m \sum_{i=k}^{n} (1 - 2p_0 - i\Delta). \tag{11}$$

Therefore, we need to ensure that

$$\sum_{i=1}^{k} i\Delta + \sum_{i=k+1}^{n} (1 - 2p_0 - i\Delta)$$
$$= \frac{k(k+1)}{2}\Delta - \frac{(k+1+n)(n-k)}{2}\Delta + (n-k)(1 - 2p_0) \geq 0. \tag{12}$$

As $p_0 \leq \frac{1}{2}, k \leq n$ simplifying Equation (12):

$$\frac{k(k+1)}{2}\Delta - \frac{(k+1+n)(n-k)}{2}\Delta + (n-k)(1 - 2p_0)$$
$$= (k^2 + k - \frac{n^2 + n}{2})\Delta + (n-k)(1 - 2p_0)$$
$$\geq (k^2 + k - \frac{n^2 + n}{2})\Delta$$

Therefore, a sufficient condition for the exploration to be effective is

$$(k^2 + k - \frac{n^2 + n}{2})\Delta \geq 0$$
$$n(n+1) \leq 2k(k+1). \tag{13}$$

In the $\mathbb{E}[R] > \frac{1}{2}$ case, the same result can be obtained similarly. When $S_1 \oplus \cdots \oplus S_n$ is an correct response, we have $p_n \geq \cdots \geq p_1 \geq p_0 \geq \frac{1}{2}$. Then $\sharp_{\text{Hit-Guided}} = m \sum_{i=0}^{n} (1 - p_i) \leq m(n+1)(1 - p_0) = \sharp_{\text{Vanilla}}$, indicating that the hit-guided strategy is worse than the vanilla strategy when using a correct guidance.

When $S_1 \oplus \cdots \oplus S_n$ is an incorrect response, under the linear hypothesis, we have $p_i = p_0 - i\Delta$. Considering the same $k$, we have for $i \leq k$, $p_i \geq \frac{1}{2}$, and for $i > k$, $p_i < \frac{1}{2}$. To make $\sharp_{\text{Hit-Guided}} \geq \sharp_{\text{Vanilla}}$, we need to ensure that

$$
\begin{aligned}
&\sharp_{\text{Hit-Guided}} - \sharp_{\text{Vanilla}} \\
&= m \sum_{i=0}^{k} (1 - p_i) + m \sum_{i=k}^{n} p_i - (n+1)m(1 - p_0) \\
&= m \sum_{i=0}^{k} i\Delta + m \sum_{i=k}^{n} (2p_0 - 1 - i\Delta) \geq 0.
\end{aligned}
\tag{14}
$$

Equation (14) is analogous to Equation (12), only differing in the sign of $1 - 2p_0$. Therefore, the condition in Equation (13) is also sufficient in the case where $p_0 > \frac{1}{2}$. Therefore, it is a sufficient condition for the hit-selection strategy in Section 3.3 to find more valid pairs. □

In the last part, we view the benefits of MARGE's exploration strategy from the viewpoint of policy optimization. We aim to show in Proposition C.3 that, when using sampled self-generated solutions as guidance, the variance of the policy gradient estimation decreases in expectation, resulting in improved policy optimization. Our analysis is based on REINFORCE for simplicity, but similar results may also be found with other optimization algorithms.

**Proposition C.3.** *Suppose $Y_0 = s_1 \oplus s_2 \oplus \cdots \oplus s_M$ are self-generated guidance solutions for all queries $q \in \mathcal{Q}$. Let $\hat{g}[\theta]$ denote the MARGE policy gradient estimator from the objective in Equation (4), conditioned on guidance solution $Y_0$; Let $\tilde{g}[\theta]$ denote the policy gradient estimator of vanilla exploration. When the number of rollouts $\tau$ is the same, the variance of the MARGE gradient estimation is lower in expectation, i.e.*

$$
\mathbb{E}_{Y_0} Var_{\tau|Y_0}[\hat{g}|Y_0] \leq Var_\tau[\tilde{g}].
\tag{15}
$$

*Proof.* For the objective in Eq 4, the vanilla policy estimator $\tilde{g}[\theta]$ can be written as

$$
\tilde{g}[\theta] = \frac{1}{|\mathcal{Q}|N} \sum_{q \in \mathcal{Q}} \sum_{i=1,\ldots,N} (r(q, y_i) - \log \pi_{\text{ref}}(y_i|q) + \log \pi_\theta(y_i|q)) \nabla_\theta \log \pi_\theta(y_i|q),
$$

where $y_i$ are randomly sampled responses from the policy, and $q$ is a query in the query set $\mathcal{Q}$. Suppose $y_i$ can be written as the concatenation of random variables at different steps $y_i = S_1 \oplus \cdots \oplus S_{M'}$. In the following part, we let $\hat{r}(q, y_i) = r(q, y_i) - \log \pi_{\text{ref}}(y_i|q) + \log \pi_\theta(y_i|q)$ for simplicity.

The policy gradient estimator $\hat{g}[\theta]$ obtained with our method with guidance $Y_0$ can be written as

$$
\hat{g}[\theta|Y_0] = \frac{1}{|\mathcal{Q}|MN} \sum_{q \in \mathcal{Q}} \sum_{i=1,\ldots,N} \sum_{j=0,\ldots,M-1} \hat{r}(q, s_0 \oplus \cdots \oplus s_j \oplus y_{ij}) \nabla_\theta \log \pi_\theta(y_{ij}|q \oplus s_0 \oplus \cdots \oplus s_j).
$$

Here, we define $s_0$ as the empty string, and $y_{ij}$ represents a response sampled to complete from step $j$.

We define a random variable that conditions on $Y_0$ as follows:

$$
\mathcal{G}^{(i)}|_{Y_0} = \frac{1}{|\mathcal{Q}|} \sum_{q \in \mathcal{Q}} \hat{r}(q, s_0 \oplus \cdots \oplus s_i \oplus y) \nabla_\theta \log \pi_\theta(y|q \oplus s_0 \oplus \cdots \oplus s_i).
$$

$\mathcal{G}^{(i)}|_{Y_0}$ represents the gradient when given query $q$ and the first $i$ steps of its corresponding guidance $Y_0$ (i.e. $s_0 \oplus \cdots \oplus s_i$). In this case, we can represent $\hat{g}[\theta|Y_0]$ and $\tilde{g}[\theta]$ with $\mathcal{G}^{(i)}$ as:

$$
\begin{aligned}
\hat{g}[\theta|Y_0] &= \frac{1}{MN} \sum_{i=1,\ldots,N} \sum_{j=0,\ldots,M-1} \mathcal{G}_i^{(j)}|_{Y_0}, \\
\tilde{g}[\theta] &= \frac{1}{MN} \sum_{i=1}^{MN} \mathcal{G}_i^{(0)}.
\end{aligned}
\tag{16}
$$

We modify the number of Monte Carlo samples $N$ to $MN$ of $\tilde{g}[\theta]$ to match the number of samples of $\hat{g}[\theta|Y_0]$. Since samples are collected independently, the variance of the policy gradient estimator can be written as:

$$\text{Var}_{\tau|Y_0}\hat{g}[\theta|Y_0] = \frac{1}{M^2N}\sum_{j=0,\ldots,M-1}\text{Var}[\mathcal{G}^{(j)}|Y_0],$$

$$\text{Var}_\tau\tilde{g}[\theta] = \frac{1}{M^2N}M\text{Var}[\mathcal{G}^{(0)}].$$

Now we want to show that $\mathbb{E}_{Y_0}\text{Var}[\mathcal{G}^{(j)}|Y_0] \leq \text{Var}[\mathcal{G}^{(0)}]$ for $j \geq 0$. Let's consider the relation between $\mathcal{G}^{(j)}|_{Y_0}$ and $\mathcal{G}^{(j+1)}|_{Y_0}$:

$$\mathcal{G}^{(j+1)}|_{Y_0} = \mathcal{G}^{(j)}|_{Y_0,S_{j+1}=s_{j+1}}.$$

By the law of total variance, we have

$$\text{Var}[\mathcal{G}^{(j)}|Y_0] = \text{Var}[\mathbb{E}_{S_{j+1}}[\mathcal{G}^{(j)}|Y_0, S_{j+1} = s_{j+1}]] + \mathbb{E}_{S_{j+1}}[\text{Var}[\mathcal{G}^{(j)}|Y_0, S_{j+1} = s_{j+1}]].$$

As variance is non-negative, we have:

$$\text{Var}[\mathcal{G}^{(j)}|Y_0] \geq \mathbb{E}_{S_{j+1}}\text{Var}[\mathcal{G}^{(j+1)}|Y_0].$$

Since the guidance solution $Y_0$ is randomly sampled from the policy's generation, its transition probability is the same as the one during Monte Carlo samples. This means completing an intermediate state from the guidance is equivalent to directly sampling from the start. Thus, we have:

$$\text{Var}[\mathcal{G}^{(j)}|Y_0] \geq \mathbb{E}_{S_0}\text{Var}[\mathcal{G}^{(1)}|Y_0] \geq \mathbb{E}_{S_0,S_1}\text{Var}[\mathcal{G}^{(2)}|Y_0] \geq \cdots \mathbb{E}_{S_0,S_1,\ldots,S_{M-1}}\text{Var}[\mathcal{G}^{(M)}|Y_0],$$

$$\text{Var}[\mathcal{G}^{(0)}] \geq \mathbb{E}_{Y_0}\text{Var}[\mathcal{G}^{(j)}|Y_0], \forall j = 0, 1, \ldots, M.$$

Therefore,

$$\mathbb{E}_{Y_0}\text{Var}_{\tau|Y_0}[\hat{g}|Y_0] \leq \text{Var}_\tau[\tilde{g}].$$

$\square$

*Remark* C.4. Based on the proof in Proposition C.3 (Equation (16)), we may even conclude that, if the guidance solution $Y_0$ is sampled from the policy, then the MARGE gradient estimation is unbiased in expectation over $Y_0$. However, as we only select one solution for each query, in our case, the bias of the MARGE estimator is not zero. Nevertheless, this issue can be mitigated by using multiple guidance solutions for each query when more computation is available.

## D. Scaling Trends of Self-Training

Based on our results of training different algorithms on Qwen2-Instruct, we find that, before performance saturation, the logarithm function $y = c_1 + c_2 \ln x$ best describes the scaling trend between consumed self-training samples and the resulting performance before performance saturation. When using the accuracy on MATH500 as an indicator, we have the fitted trends shown in Tab. 7 and Fig. 6. The metrics above clearly showcase the effectiveness of MARGE in improving scaling training data. The other algorithms are excluded as they saturate too fast to gain enough datapoints for regression.

Table 7. The fitted coefficients for different methods

|  | $c_1$ | $c_2$ |
|---|---|---|
| MARGE | 53.05 | 2.287 |
| GRPO | 52.92 | 1.302 |
| SFT(RFT) | 54.89 | 0.99 |

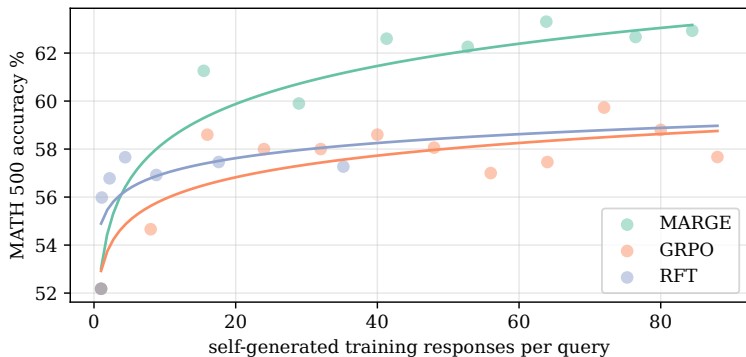

*Figure 6.* The fitted line and datapoints for different methods

# E. Prompts

For evaluation, we adopt their default chat template for Qwen2-7B-Instruct, Qwen2.5-7B-Instruct, and Llama3.1-8B-Instruct. The prompt is adopted as follows.

> **Prompt for Evaluation**
>
> {Question}
> Please reason step by step, and put your final answer within \boxed{}.

For MetaMath-Mistral-7B, we use the same Alpaca format as their official repo[4].

> **Prompt for Evaluating MetaMath-Mistral-7B model**
>
> Below is an instruction that describes a task. Write a response that appropriately completes the request.
>
> ### Instruction:{Question}
> Please reason step by step, and put your final answer within \boxed{}.
>
> ### Response: Let's think step by step.

During training, we divide a guidance solution into multiple reasoning steps by identifying special tokens or counting token number. In the case of identifying special delimiters like `"Step i:"`, we need to ensure the model outputs in a structured way. Here, we add a CoT prefix `"Let's think step by step.\n Step 1:"` to the model's output. It ensures the output follows the format and contains the delimiters required. We apply this technique to the Qwen2-7B-Instruct and Qwen2.5-7B-Instruct models.

# F. Case Study

In this section, we provide an illustrative example of how MARGE works with Qwen2-7B-Instruct in Figure 7 and with Llama3.1-8B-Instruct in Figure 8.

# G. Failure Analysis

Based on our theoretical analysis and empirical results above, MARGE is capable of improving the exploration process and finally the reasoning ability of LLM. However, there are still outlier cases when the model fails to accomplish such goals. Here, we focus on two typical types of failure, from the experiments on Qwen2-7B-Instruct:

---

[4]https://huggingface.co/meta-math/MetaMath-Mistral-7B

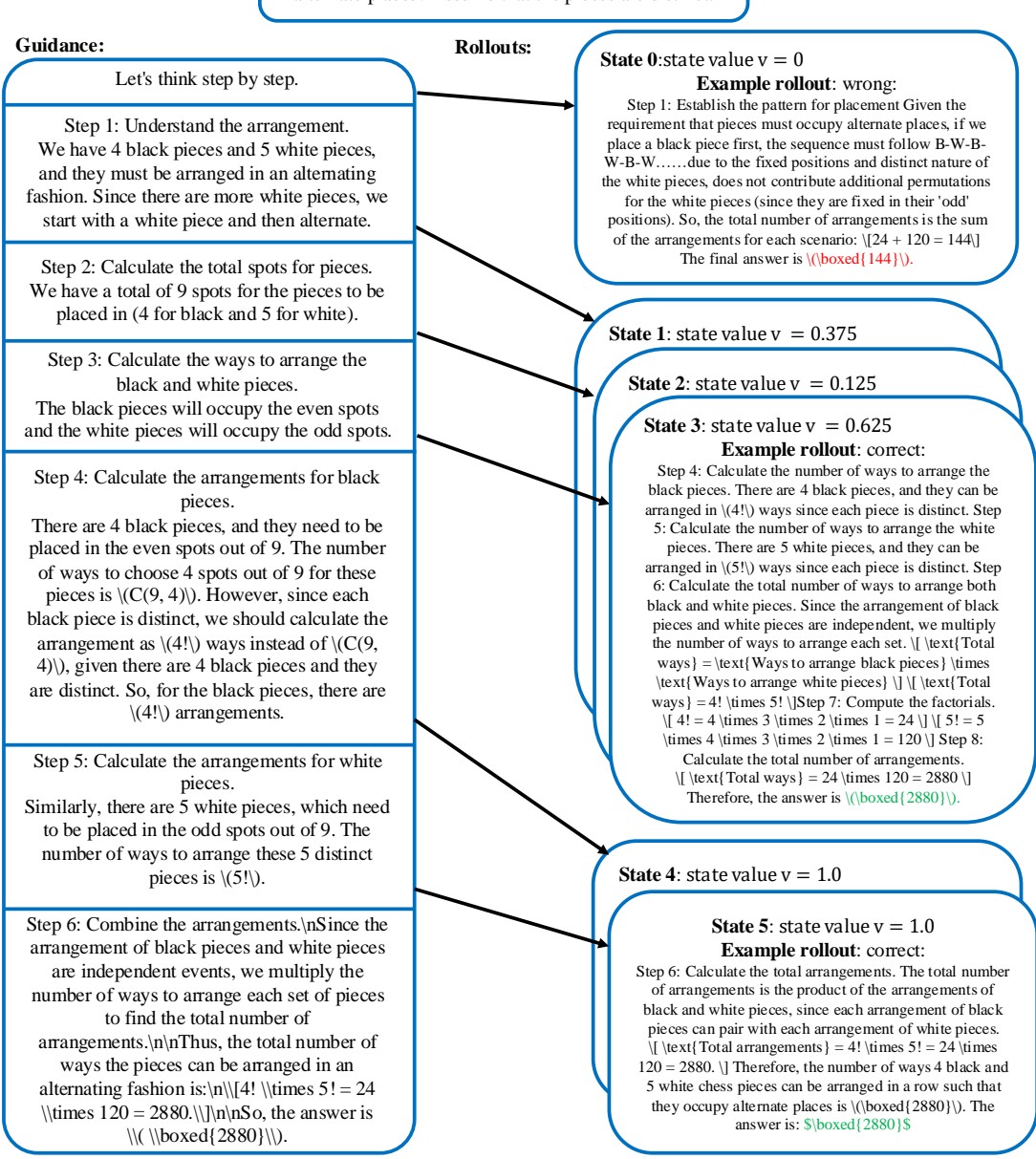

*Figure 7.* An illustrative example of how MARGE works with Qwen2-7B-Instruct. In this example, we use the delimiter `"Step i:"` to divide the guidance solution into 6 reasoning steps. In particular, to make sure the generated responses comply with the delimiter format, we add the CoT prefix for generation on all states. The arrows connect intermediate states with the exploration process from that state, including generated responses that complete the prefixes and corresponding value estimation. Of the 6 intermediate states in the guidance hit, we visualize 3 states in detail, showcasing a randomly selected rollout at this state. On each state, we sample 8 responses to complete it and estimate the state value. We display its estimated value and a randomly selected rollout as an example. The example in state 0 is partially omitted for a better visualization.

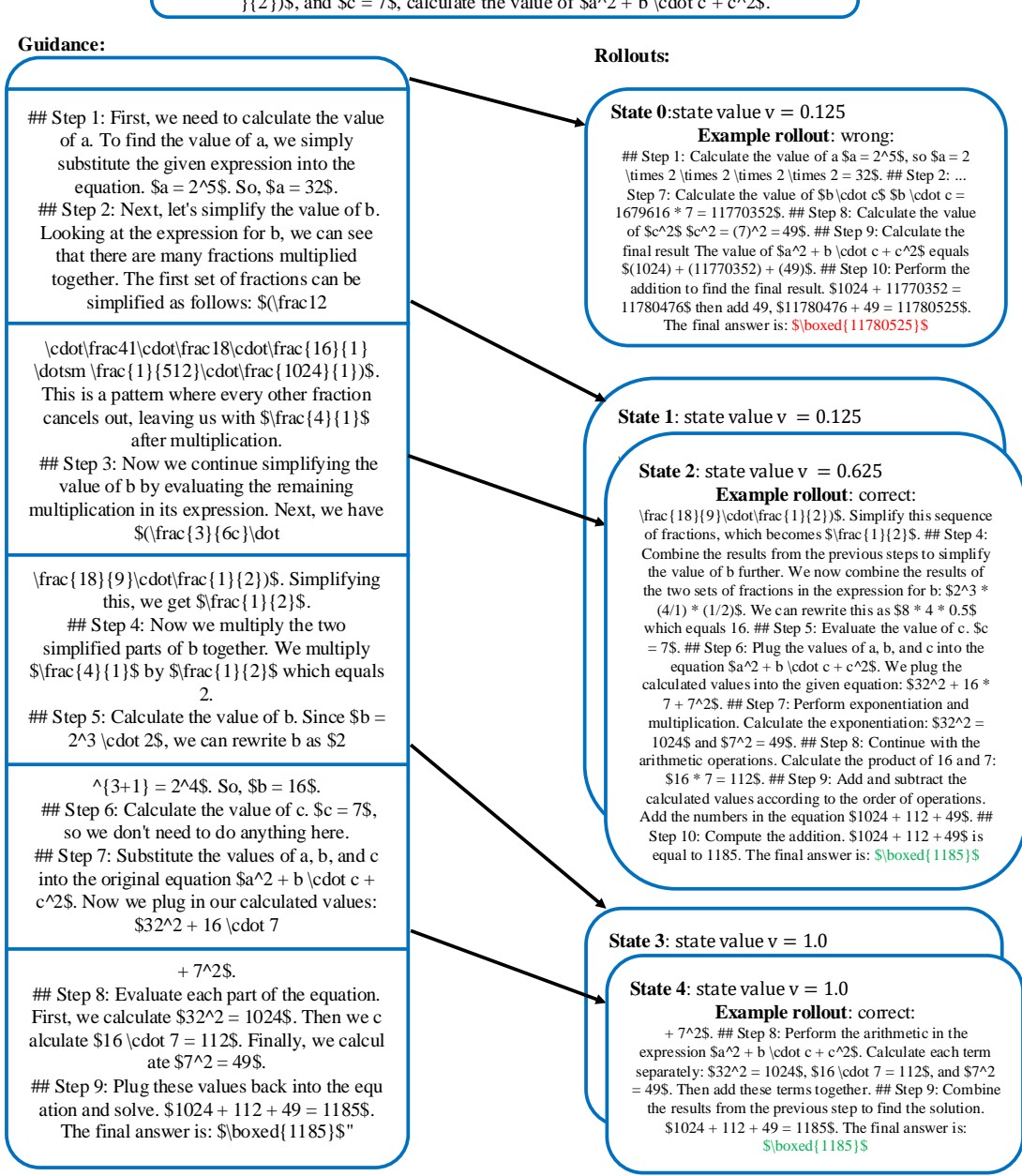

*Figure 8.* An illustrative example of how MARGE works with Llama3.1-8B-Instruct. In this example, we split the guidance based on the total token number. We derive the intermediate states such that the token number in each intermediate state forms an arithmetic sequence. The arrows connect intermediate states with the exploration process from that state, including generated responses that complete the prefixes and corresponding value estimation. Of the 5 intermediate states in the guidance hit, we visualize 3 states in detail, showcasing a randomly selected rollout at this state. On each state, we sample 8 responses to complete it and estimate the state value. We display its estimated value and a randomly selected rollout as an example. The example in state 0 is partially omitted for a better visualization.

1. queries that our trained model failed to answer;

2. cases where our model failed to find more preference pairs.

**Wrong Solutions**  We select a sample from the MATH test set, which the trained Qwen2-7B-Instruct checkpoint failed to solve. We mark wrong reasoning with red.

---

A Failed Case from MATH

**Question:** Joe was in charge of lights for a dance. The red light blinks every two seconds, the yellow light every three seconds, and the blue light every five seconds. If we include the very beginning and very end of the dance, how many times during a seven minute dance will all the lights come on at the same time? (Assume that all three lights blink simultaneously at the very beginning of the dance.)

**Correct Solution:**  The three lights blink simultaneously $t$ seconds after the start of the dance if and only if $t$ is a common multiple of 2, 3, and 5. Recall the common multiples of a set of integers are precisely the multiples of the least common multiple. Since 2, 3, and 5 are relatively prime, their least common multiple is $2 \cdot 3 \cdot 5 = 30$. Thus the light blinks $t$ seconds after the beginning of the song for $t = 0, 1, 2, \ldots, 14$, and after 14 thirty-second periods, the song ends. Thus the lights blink in unison a total of $\boxed{15}$ times.

**Model Output:**  To find out how many times all three lights (red, yellow, and blue) come on at the same time during seven-minute dance, we need to determine the least common multiple (LCM) of the blinking intervals of the three lights: 2 seconds, 3 seconds, and 5 seconds. This LCM will give us the interval at which all three lights blink simultaneously.
1. **Prime Factorization:**
- $2 = 2$
- $3 = 3$
- $5 = 5$
2. **Calculate LCM:** The LCM of a set of numbers is found by taking the highest power of all prime numbers appearing in the factorization of these numbers.
- The prime numbers involved are 2, 3, and 5.
- The highest power of each prime number in the factorizations is $2^1$, $3^1$, and $5^1$.
Therefore, the LCM is $2^1 \times 3^1 \times 5^1 = 2 \times 3 \times 5 = 30$.
This means all three lights blink simultaneously every 30 seconds.
3. **Convert Dance Duration to Seconds:** A seven-minute dance lasts $7 \times 60 = 420$ seconds.
4. **Calculate Number of Simultaneous Blinks:** To find out how many times all three lights blink simultaneously during the 420 seconds, divide the total time by the interval:

$$\frac{420}{30} = 14$$

Thus, all three lights come on at the same time **14 times** during the seven-minute dance.
The final answer is $\boxed{14}$.

---

**Invalid Guided Exploration**  As discussed in Appendix C, MARGE may not discover more valid training data when scaling generation unless the requirements are satisfied, especially when the intermediate state values increase (or decrease) too fast. Here, we showcase a query, along with its intermediate states, that fails to gain the exploration bonus because of this. As demonstrated, this is caused by the fact that many later states have state values of 1 (or 0). Designing mechanisms to filter out ineffective guidance or unnecessary states can potentially improve this framework.

---

**Question:** In a zoo there are 30 penguins, 22 zebras, and 8 tigers with some zookeepers. If the total number of heads are 132 less than the number of feet, the number of zookeepers is

---

**State 0:** (state value:0.125) Let's think step by step.
Step 1:
**State 1:** (state value:0.375) Understanding the information given:
- There are 30 penguins. Penguins have 2 feet each.
- There are 22 zebras. Zebras have 4 feet each.
- There are 8 tigers. Tigers have 4 feet each.
- There are some zookeepers, but we don't know how many yet. Let's denote the number of zookeepers as $Z$.
Step 2:
**State 2:** (state value: 1) Calculating the total number of feet:
Each animal and each zookeeper has 2 feet each. So, the total number of feet is:

$$\text{Total feet} = (30 \times 2) + (22 \times 4) + (8 \times 4) + (Z \times 2)$$

$$\text{Total feet} = 60 + 88 + 32 + 2Z$$

$$\text{Total feet} = 180 + 2Z$$

Step 3:
**State 3:** (state value: 1) Calculating the total number of heads:
Each animal and each zookeeper has 1 head. So, the total number of heads is:

$$\text{Total heads} = 30 + 22 + 8 + Z$$

$$\text{Total heads} = 60 + Z$$

Step 4:
**State 4:** (state value: 1) Using the given information:
We're told that the total number of heads is 132 less than the number of feet. This gives us the equation:

$$\text{Total heads} + 132 = \text{Total feet}$$

$$60 + Z + 132 = 180 + 2Z$$

Step 5:

