# OpenReview forum: "MARGE: Improving Math Reasoning with Guided Exploration"
_ICML.cc/2025/Conference — ICML 2025 poster_

### Official Review · Reviewer_QKKW · 2025-03-12

**Overall Recommendation:** 3

**Summary:**

This paper introduces MARGE, a framework for guided exploration in LLM mathematical reasoning for self-training pipeline.
It uses solution-guided exploration with MCTS and RL to find high-quality data, resulting in better exploration and credit assignment.
Specifically,
for each question it has a response, use rule based \n to split the response into states then from each state to finish the response and collect correct/incorrect labels, and then estimate state values based on labels, then train the policy using RL and update response.
In experiment using different LLMs and on different math dataset, they show it improves accuracy compare to baselines.

**Claims And Evidence:**

Yes

**Essential References Not Discussed:**

NA

**Experimental Designs Or Analyses:**

Yes, I checked the Table 1 main result.
The accuracy of Qwen2.5-7B-Instruct experiment on MATH is 75.3 and the MARGE improves to 76.74.
I think the improvement is not that significant. However, the results on MATH500 with pass@64 does show significant improvements over the baseline (85 > 79).
My main concern is the lack of experiments using Qwen2.5 Math based model - would this lead to better accuracy or not?
In addition, the Table1 does not show any reference results from previous literature (7B model such as PRIME-EURUS, ACEMATH, rStar), it only compares the results under its own setting, which makes it hard for reader to understand how this method compares to the current literature and state of the art results at 7B level.

============ Post rebuttal ============

Author has added additional Qwen-math results and add references to other models. They have addressed my concerns and thanks for the effort.

**Methods And Evaluation Criteria:**

Yes

**Other Comments Or Suggestions:**

the definition of state for implementation seems quite simple: based on "step i" or based on token count...What would be a better way to separate the reasoning process into different steps if compute is not a constraint?

**Other Strengths And Weaknesses:**

Strength: novel framework
Weaknesses: weak result

**Questions For Authors:**

NA

**Relation To Broader Scientific Literature:**

This contributes to math LLM RL using MCTS to provide state level supervision.

**Theoretical Claims:**

N/A

---

> ### Author Rebuttal · Authors · 2025-04-01
>
> Thank you for your thorough review! We are happy to know you found MARGE to be novel and contribute to math LLM RL. Here, we appreciate the chance to address your questions.
>
> # W1: lack of experiments using Qwen2.5 Math based model:
>
> Thank you for your advice! To demonstrate MARGE's effectiveness on state-of-the-art math models, we conducted new experiments using **Qwen2.5-Math-7B-Instruct**. As models like this often undergo extensive and successful RL post-training, achieving further gains can be challenging[1], making this a good test case to showcase MARGE's effects. Due to its high reasoning ability, we randomly sample ~14k queries from the BigMath[2] dataset as training queries. Results (average of 3 runs) are below. We didn't test pass@64 on MATH due to its large size.
>
> **Table 1**:Performance Comparison on Math Benchmarks  Pass@1 / Pass@64 Accuracy (%)
>
> | Pass1/64 Acc (%)         | MATH@1 | MATH500       | OlympiadBench | CollegeMath   |
> | ------------------------ | ------ | ------------- | ------------- | ------------- |
> | Qwen2.5-Math-7B-Instruct | 83.48  | 83.33 / 86.40 | 40.80 / 48.64 | 46.95 / 48.62 |
> | PPO                      | 83.37  | 83.26 / 86.12 | 40.75 / 47.14 | 47.05 / 48.63 |
> | **MARGE (Ours)**         | 84.46  | 85.04 / 89.92 | 41.58 / 49.49 | 47.40 / 49.02 |
> | ACEMath                  | 83.13  | 83.42 / 85.72 | 42.76 / 50.32 | 48.68 / 50.45 |
> | PRIME-EURUS              | 80.08  | 80.70 / 88.58 | 40.99 / 58.96 | 48.22 / 51.97 |
>
> These results show MARGE effectively improves performance even on strong, instruction-tuned models. Notably, MARGE also yields larger relative gains on pass@k compared to pass@1, underscoring its benefit in enhancing **exploration diversity** and finding a wider range of correct solutions.
>
> # W2: no reference results included
>
> Thank you for pointing this out. Our initial tables focused on **controlled** comparisons within our experimental setup for clarity. We agree context is valuable and have now included reference results from recent literature (AceMath, PRIME-EURUS) in Table 1 above for comparison. We additionally test their pass@64 with open-source models. We will add them together in our revised paper.
>
> What's more, we also want to emphasize MARGE's specific contribution: **improving exploration efficiency** to enable **scaled self-training**, reflected in both pass@1 and pass@k gains. This contribution is complementary to concurrent methods like PRIME-EURUS (implicit rewards modeling), and AceMath (curated data/process supervision). rStar proposes a system for SLMs to reason and resembles the idea of test-time scaling, and we cited it in related works. MARGE enhances the underlying exploration process in RL and can potentially be combined with these approaches, not as counterparts, for future enhancement in LLM reasoning.
>
> # Question 1: Other ways to separate the reasoning process
>
> We select these ways due to both their simplicity and their effectiveness. We also tested using "\n\n", which indicates new paragraphs in latex syntax, to separate the reasoning process. This yields more intermediate states than `Step i` or token counts but does not result in better final performance. Therefore, we believe when computation is not a constraint, simply more states aren't always better.
>
> Alternative segmentation strategies are indeed interesting future directions for exploration and process-based rewards research. Approaches could include:
> - Rule-based segmentation using logical structure (e.g., identifying equations, logical connectors), but it should also match the model's output style.
> - Using an auxiliary LLM to identify meaningful intermediate reasoning steps.
>
> Overall, exploring better segmentation is an interesting problem for not only our work but also the development of process rewards and PRMs, and thus can be further researched in the future.
>
> We hope these clarifications and additional results address your concerns! We believe MARGE contributes to enhancing LLM reasoning through improved exploration.
>
> [1] Gao, Jiaxuan, et al. "On designing effective rl reward at training time for llm reasoning." arXiv preprint arXiv:2410.15115 (2024).
>
> [2] Albalak, Alon, et al. "Big-Math: A Large-Scale, High-Quality Math Dataset for Reinforcement Learning in Language Models." arXiv preprint arXiv:2502.17387 (2025).

---

### Official Review · Reviewer_8oFv · 2025-03-13

**Overall Recommendation:** 3

**Summary:**

The paper presents MARGE, a method that improves the self-training of Large Language Models (LLMs) in math reasoning. MARGE relies on “guided exploration,” reusing partial solutions (“hits”)—correct or incorrect—to fix shared prefixes while varying subsequent steps. This stabilizes the generation of positive and negative examples in multi-step math tasks. Experiments show significant gains over baselines (SFT, PPO, DPO) on benchmarks like MATH, GSM8K, and OlympiadBench, with improved pass@1 and pass@k accuracy. ## update after rebuttal

**Claims And Evidence:**

Yes

**Essential References Not Discussed:**

No

**Experimental Designs Or Analyses:**

Yes

**Methods And Evaluation Criteria:**

Yes

**Other Comments Or Suggestions:**

No

**Other Strengths And Weaknesses:**

**Strengths**
- Directly addresses spurious correlations in self-generated data.
- Demonstrates clear empirical gains over various baselines.
- Provides extensive ablations and theoretical insights on why guided exploration yields richer training data.

Weaknesses:
- The method of choosing hit selection is still ambiguous in cases where no correct or wrong answers are generated. Due to its high frequency, it is not trivial to skip this point, especially for cases with no correct generated answers where exploration should be promoted.
- The statement “This solution is based on the simple intuition of increasing the likelihood of finding the right answer to a difficult question and possible failure cases to an easy one” does not hold all the time. If we start from a wrong intermediate point, no matter how the latter process is, we still get a wrong answer, right? It is a bit confusing why you chose the failed case as the hit. If the failure point lies in the first step, does it mean that all generated roll-outs are wrong?

**Questions For Authors:**

- Do you assume that if the final answer is correct, then the whole process is correct? If yes, is it in conflict with Proposition C.1 at the point  “The reward function that gives 1 if and only if \( S_1 \oplus \cdots \oplus S_n \) is a correct solution to \( q \).”
- How do you handle queries where no suitable correct or incorrect “hit” is found?

**Relation To Broader Scientific Literature:**

The paper relates to the training of reasoning capability of LLMs.

**Theoretical Claims:**

Yes

---

> ### Author Rebuttal · Authors · 2025-04-01
>
> Thank you for your valuable review! We are glad to learn that you find MARGE to be effective, contains empirical gains, and has extensive ablations and theories. We appreciate the opportunity to clarify the points raised.
>
> # W1: choosing hit selection in cases where no correct or wrong answers are generated
>
> Thank you for your question regarding the hit selection method.
>
> 1. Cases with No Correct Answers Generated
>
> We agree with your insightful point that promoting exploration, especially when no correct answers are initially found, is crucial. We believe this question contains two key points: at least get one answer correct, and decrease the difficulty to find them.
>
> The first key point remains a fundamental challenge, dependent on both exploration strategies and the backbone model's capabilities. Our current approach, consistent with recent reasoning works[2,3], is to increase the number of i.i.d. samples generated per query to improve the probability of finding a correct trajectory. Empirical results on Qwen2 demonstrate:
> - At 32 samples: ~300/8888 questions lack correct answers
> - At 256 samples: only 20 questions remain without valid answers
>
> We remove remaining unsolved cases from training, leaving them to more capable models.
>
> For the second point, as shown in Fig. 5, once a valid answer is sampled, MARGE greatly improves the exploration efficiency and surpasses baselines.
>
> 2. Cases with No Incorrect Answers
> 	- This indicates problems are "too easy" for the current model
> 	- We exclude them from training as they provide limited learning value, and focus more on challenging cases.
>
> # W2: failed cases as the guidance hits
>
> We appreciate your concern about our rationale for using incorrect responses as guidance hits for easier questions. Here we answer your questions and clarify our motivation:
>
> *If we start from a wrong intermediate point, no matter how the latter process is, we still get a wrong answer*
>
> We respectfully disagree based on our findings and concurrent works[1,2]. LLMs possess the potential to recover from wrong intermediate points and reach a correct final answer. Exploring data to reinforce such ability is a key goal of MARGE. We quantify this effect as follows.
>
> As it is hard to decide the wrong states, we suppose states, where the estimated state value drops the most, are the false ones. We count the ratio of these states that could be recovered from, and the accuracies when completing them. We also count them when failures are at the first step.
>
> | | recoverable wrong step % | Acc % of recovery | recoverable wrong first step % | Acc % of recovery (first step) |
> | -- | -- | -- | -- | -- |
> | Qwen2 | 73 | 44 | 83 | 54 |
> | Llama3.1 | 74 | 36 | 88 | 46 |
> | MetaMath | 51 | 21 | 73 | 33|
> | Qwen2.5 (different queries) | 65| 35 | 80 | 62 |
>
> *If the failure point lies in the first step, does it mean that all generated roll-outs are wrong?*
>
> No, as the table above demonstrates (columns 3 and 4), models can often recover even when the error occurs at the very first step.
>
> While wrong intermediate states do decrease the expected accuracy (as implied by the red line in Fig. 2), the possibility of recovery exists and is valuable. By selecting failed cases as negative guidance hits, we increase the portion of wrong rollouts for easy problems. These rollouts provide valuable learning targets for models' robustness:
> - Showcase common mistakes of models,
> - Demonstrate ways to avoid or even correct errors.
>
> We validate the effectiveness of selecting wrong hits as guidance for exploration (Fig. 5) and final results (Tab. 3). Therefore, we believe this design choice is well-motivated and experimentally supported.
>
> # Q1: Assumption about the whole process is correct
>
> No, we don't assume this. Here, we want to express that the reward (in Prop C.1) is 1 iff the final answer given by $(S_1\oplus\cdots\oplus S_n)$ is correct. Trajectories with corrected intermediate errors are valid positive examples and help improve the model's reasoning abilities.
>
> # Q2: Handling queries without suitable hits found
>
> We progressively sample more responses for queries with no suitable hits found, as discussed detailedly in W1. Currently, we sample up to 256 responses. We discard the queries if they still get no suitable hits, the same as concurrent works[2,3]. Once a suitable hit is found, MARGE greatly improves exploration.
>
> We believe MARGE contributes to enhancing LLM reasoning through improved exploration. We hope this rebuttal clarifies our method and addresses your concerns that led to your decision not to recommend acceptance of our work. We look forward to your feedback!
>
> [1]Jaech, Aaron, et al. "Openai o1 system card." arXiv preprint arXiv:2412.16720 (2024).
>
> [2]Guo, Daya, et al. "Deepseek-r1: Incentivizing reasoning capability in llms via reinforcement learning." arXiv preprint arXiv:2501.12948 (2025).
>
> [3]Cui, Ganqu, et al. "Process reinforcement through implicit rewards." arXiv preprint arXiv:2502.01456 (2025).

---

### Official Review · Reviewer_SZQi · 2025-03-13

**Overall Recommendation:** 4

**Summary:**

The paper introduces MARGE (Math Reasoning with Guided Exploration), a framework to enhance mathematical reasoning in Large Language Models (LLMs). It addresses two fundamental challenges in LLM reasoning: the scarcity of high-quality training data and the difficulty of exploring reasoning paths effectively. Unlike traditional self-training approaches, which often suffer from spurious correlations in self-generated datasets, MARGE introduces guided exploration to improve data diversity and reasoning accuracy. The core idea is hit-guided exploration, where intermediate reasoning states from self-generated solutions are systematically explored, leading to improved credit assignment and scalability in training. The paper provides extensive experimental results showing that MARGE improves both single-shot accuracy (pass@1) and exploration diversity (pass@k) across multiple benchmarks and backbone models.

The highlight to me is that the method can keep the reward on policy compared to other pretrained PRM, yet the problem is the on-line computation cost.

**Claims And Evidence:**

Some of the statements are supported but may not be enough.

Based on the numbers presented in the tables, the gain of MARGE is not convincingly significant. And the computation cost was not fully discussed in the paper.

**Essential References Not Discussed:**

Multi-Step Problem Solving Through A Verifier: An Empirical Analysis on Model-Induced Process Supervision, EMNLP 2024
Improve mathematical reasoning in language models by automated process supervision, 2024

both papers has similar ideas as Math-Shepherd of using monte carlo estimation (hit-guide) to rate the partial solutions for math problems. The cost of running the experiments are discussed in those papers.

**Experimental Designs Or Analyses:**

The design of experiments are valid.

**Methods And Evaluation Criteria:**

The method and evaluation benchmarks, metrics are reasonable.

The benchmarks are mostly math but pretty comprehensive.

**Other Comments Or Suggestions:**

No

**Other Strengths And Weaknesses:**

Strength:
1. The method keeps the reward on-policy which is critical for process supervision.

Weakness:
1. This method will increase the computation, yet not fully discussed in the paper.

**Questions For Authors:**

No

**Relation To Broader Scientific Literature:**

The paper builds on prior work in LLM self-training and reinforcement learning for reasoning.

**Theoretical Claims:**

The Proof MERGA in the Appendix C to approve the method unbiased is correct.

---

> ### Author Rebuttal · Authors · 2025-04-01
>
> Thank you for your helpful review! We are glad to learn you find MARGE to be novel, keep reward on-policy, and be reasonably evaluated. Here, we appreciate the chance to address your questions.
>
> # Claims and Evidence: not convincingly significant
>
> We conducted experiments with models of different abilities to demonstrate the effectiveness of our method. MARGE not only increases the pass@1 accuracy of models. More interestingly, it is capable of significantly increasing pass@k accuracy and improving the model's diversity, which is a trend not seen in most post-training methods.
>
> We conduct new experiments using **Qwen2.5-Math-7B-Instruct**. As models like this often undergo extensive and successful RL post-training, achieving further gains can be challenging[1], making this a good test case to showcase MARGE's effects. Due to its high reasoning ability, we randomly sample ~14k queries from the BigMath[2] dataset as training queries. Results (average of 3 runs) are below. We didn't test pass@64 on MATH due to its large size.
>
> **Table 1**:Performance Comparison on Math Benchmarks  Pass@1 / Pass@64 Accuracy (%)
>
> | Pass1/64 Acc (%)         | MATH@1 | MATH500       | OlympiadBench | CollegeMath   |
> | ------------------------ | ------ | ------------- | ------------- | ------------- |
> | Qwen2.5-Math-7B-Instruct | 83.48  | 83.33 / 86.40 | 40.80 / 48.64 | 46.95 / 48.62 |
> | PPO                      | 83.37  | 83.26 / 86.12 | 40.75 / 47.14 | 47.05 / 48.63 |
> | **MARGE (Ours)**         | 84.46  | 85.04 / 89.92 | 41.58 / 49.49 | 47.40 / 49.02 |
>
> These results show MARGE effectively improves performance even on strong, instruction-tuned models. Besides the improvement on pass@1, MARGE also yields large improvements on pass@k, indicating its benefit in enhancing **exploration diversity** and finding a wider range of correct solutions.
>
> # Missing references:
>
> Thank you for your kind reminder of these works! We will include these two works in the first part of our related works, where we discuss process supervision methods in LLM reasoning.
>
> # W1: computation cost
>
> Thank you for your advice! Adding a discussion on computation cost is important for the integrity and rigor of our work, and we will include this part in the revised version.
>
> Compared to vanilla ways (DPO, SFT, PPO, ...), our method increases the number of prompts to generate when other parameters are controlled. We argue that MARGE only changes the coefficient of the time complexity but **not its asymptotic behavior**, thus is acceptable in practice. This coefficient is determined by the number of intermediate states for each query. In our experiment results, it is ~3.3 on Qwen2, Llama3.1, and MetaMath where there are about 5 states per query, and ~4.9 on Qwen2.5 with about 8 states per query.
>
> Possible ways to reduce the computation cost of our method also exist, like removing unnecessary states from the Monte Carlo estimation. We believe this can be an interesting topic for future works.
>
> Here, in Tab. 2, we present the results of MARGE and some baselines on Qwen2 when MARGE uses **less** computation, such that the training GPU time is roughly the same. In such cases, our method still exhibits advantages over baselines. We compare the results when baselines utilize more computation in Tab. 2 of our paper.
>
> **Table2:**
>
> | Acc %     | MATH  | GSM8k | CollegeMATH | OlympiadBench |
> | --------- | ----- | ----- | ----------- | ------------- |
> | PPO       | 58.7  | 88.47 | 35.72       | 21.82         |
> | REINFORCE | 59.81 | 88.32 | 35.58       | 24.49         |
> | MARGE     | 60.67 | 88.10 | 35.81       | 25.28         |
>
> Here, we want to emphasize that, while our method utilizes more generation computation, it is our **goal and contribution** to **scale up** the computation to make the most use of the current query set. High-quality problems are getting harder to acquire. Therefore, we develop MARGE, with stronger exploration ability, to find more high-quality training samples.
>
> As we demonstrate in Tab. 2 of our paper, adding more computation for baselines results in overfitting and degradation in performance. As recent progress in LLM inference, we believe adding up computation to automatically improve models is becoming the more feasible and promising way, highlighting the contribution of MARGE.
>
> We believe MARGE contributes to enhancing LLM reasoning through improved exploration. We hope these clarifications and additional results address your concerns!
>
> [1] Gao, Jiaxuan, et al. "On designing effective rl reward at training time for llm reasoning." arXiv preprint arXiv:2410.15115 (2024).
>
> [2] Albalak, Alon, et al. "Big-Math: A Large-Scale, High-Quality Math Dataset for Reinforcement Learning in Language Models." arXiv preprint arXiv:2502.17387 (2025).

---

### Official Review · Reviewer_GXc7 · 2025-03-14

**Overall Recommendation:** 4

**Summary:**

The paper introduces a hit-guided exploration method to enhance LLMs’ mathematical reasoning by systematically exploring intermediate reasoning states. Using Monte Carlo simulations for better credit assignment, MARGE improves accuracy and reasoning diversity across multiple benchmarks without needing extra value models, making self-training more effective.

**Claims And Evidence:**

Yes. Experiments support most claims in the paper.

**Essential References Not Discussed:**

N/A

**Experimental Designs Or Analyses:**

Yes
The main result is comprehensive and reasonable.
Ablation study looks reasonable.

**Methods And Evaluation Criteria:**

Yes. Datasets are proper to the task.

**Other Comments Or Suggestions:**

1. The caption of Fig. 2 needs more explanation.

**Other Strengths And Weaknesses:**

Strengths:
1. The paper compares against the most popular RL methods on widely used models.
2. The ablation studies confirm that hit selection strategy and guidance updates significantly impact performance.

Weaknesses:
1. The paper lacks failure analyses.
2. The paper claims MARGE enables scaling self-generated responses more effectively but does not show an explicit scaling trend.

**Questions For Authors:**

1. MARGE avoids additional value models, but how does its computational cost compare to DPO, PPO, or GRPO?
2. How many Monte Carlo samples (n) are used for value estimation (Eq. 2)? How does it relate to training efficiency?

**Relation To Broader Scientific Literature:**

The paper differs from most previous RL methods.
The hit-guided exploration strategy is well-motivated, ensuring better coverage of reasoning steps.
The use of Monte Carlo simulation is widely used in many methods and is computationally efficient.
Iterative guidance updates ensure on-policy data generation, improving model alignment.

**Theoretical Claims:**

It is in appendix C.

---

> ### Author Rebuttal · Authors · 2025-04-01
>
> Thank you for your valuable reviews! We are more than happy to learn that you find our method to be well-motivated, contain reasonable and comprehensive results. Here we appreciate the chance to address your concerns.
>
> # W1: lacks failure analyses
>
> Thank you for your advice to improve our work! We also believe adding failure analyses will further enhance our work. Here, we plan to add failure analyses on:
>
> 1. special queries that our trained model failed to answer;
> 2. special cases where our model failed to find more preference pairs.
>
> However, due to the limitation on the rebuttal length, we are unable to provide you full examples here. We will add this part as a chapter in the appendix in the updated version.
>
> # W2: does not show an explicit scaling trend
>
> Thank you for your advice! An explicit trend can better showcase MARGE's improvement and we will update it in the revised paper. Based on our data points on Qwen2, we find the logarithm function $y=c_1+c_2\ln(x)$ best fits the scaling trend between MATH500 accuracy $y$ and number of training samples $x$. We find the following coefficients for different methods:
>
> - MARGE: $c_1=53.05, c_2=2.287$;
> - GRPO: $c_1=52.92, c_2=1.302$;
> - RFT: $c_1=54.89, c_2=0.99$.
>
> The metrics above clearly showcase the effectiveness of MARGE in scaling training data. We plot the data points as well as fitted scaling lines in the figure [(link)](https://anonymous.4open.science/r/MARGE-ACE8/explicit_scaling_trend.png). We only include three lines for clarity of the picture, and we will include the results of other algorithms in the paper.
>
> # Comment
>
> Thank you for your suggestion! More explanation here will make our paper clearer. We will update the caption of Fig. 2 to the following:
>
> > Average accuracies when starting from different intermediate states of correct solutions (blue) and incorrect ones (red) with Qwen2-7B-Instruct. A larger state index indicates being closer to the end. On average, completing from a correct (incorrect) state increases the portion of correct (incorrect) answers, which boosts the exploration of more training data.
>
> # Q1: computation cost
> Thank you for your advice! Adding a discussion on computation cost is important for the integrity and rigor of our work, and we will include this part in the revised version.
>
> Compared to vanilla ways (DPO, SFT, PPO, ...), our method increases the number of prompts to generate when other parameters are controlled. We argue that MARGE only changes the coefficient of the time complexity but **not its asymptotic behavior**, thus is acceptable in practice. This coefficient is determined by the number of intermediate states for each query. In our experiment results, it is ~3.3 on Qwen2, Llama3.1, and MetaMath where there are about 5 states per query, and ~4.9 on Qwen2.5 with about 8 states per query.
>
> Possible ways to reduce the computation cost of our method also exist, like removing unnecessary states from the Monte Carlo estimation. We believe this can be an interesting topic for future works.
>
> In Tab. 2, we present the results of MARGE and some baselines on Qwen2 when MARGE uses **less** computation, such that the training GPU time is **roughly the same**. In such cases, our method still exhibits advantages over baselines.
>
> **Table2:**
>
> | Acc %     | MATH  | GSM8k | CollegeMATH | OlympiadBench |
> | --------- | ----- | ----- | ----------- | ------------- |
> | PPO       | 58.7  | 88.47 | 35.72       | 21.82         |
> | REINFORCE | 59.81 | 88.32 | 35.58       | 24.49         |
> | MARGE     | 60.67 | 88.10 | 35.81       | 25.28         |
>
> Here, we want to emphasize that, while our method utilizes more generation computation, it is our **goal and contribution** to **scale up** the computation to make the most use of the current query set. High-quality problems are getting harder to acquire. Therefore, we develop MARGE, with stronger exploration ability, to find more high-quality training samples.
>
> As we demonstrate in Tab. 2 of our paper, adding more computation for baselines results in overfitting and degradation in performance. As recent progress in LLM inference, we believe adding up computation to automatically improve models is becoming the more feasible and promising way, highlighting the contribution of MARGE.
>
> # Q2
>
> Here we use n=8 samples for value estimation. Higher Monte Carlo samples (n) provides more accurate value estimation and may yield better results. But at the same time, it leads to a linear increase in the number of generated tokens, decreasing training efficiency. We choose n=8 to balance this two effects.
>
> We hope these clarifications and additional illustrations address your concerns!

---

### Decision · Program_Chairs · 2025-05-01

**Decision:**

Accept (poster)

**Comment:**

The paper proposes MARGE (Math Reasoning with Guided Exploration), a method designed to enhance mathematical reasoning in large language models (LLMs). It addresses two challenges in LLM reasoning: the scarcity of high-quality training data and the difficulty of effectively exploring reasoning paths. In contrast with traditional self-training approaches, which often suffer from spurious correlations in self-generated datasets, the paper method employs hit-guided exploration to improve both data diversity and reasoning accuracy. The core idea involves  hit-guided exploration, where intermediate reasoning states from self-generated solutions are systematically explored, leading to improved credit assignment. Experiments demonstrate that MARGE improves single-shot accuracy (pass@1) and exploration diversity (pass@k) across benchmarks and backbone models.
An advantage of MARGE is its ability to maintain on-policy rewards, unlike other pre-trained PRM methods. However, this comes at the increased computation cost.

The reviewers highlighted several strengths of the paper, including its comprehensive experiments comparing MARGE against popular RL methods on widely used models, as well as ablation studies confirming that hit selection strategy and guidance updates improve performance. The paper provides theoretical insights into why the hit guided exploration yields richer training data. The method addresses spurious correlations in self-generated data, demonstrates  empirical improvements over baselines.

However, some weaknesses were initially noted in the reviews. These included a lack of failure analysis, insufficient evidence for MARGE’s scalability in self-generating responses (no explicit scaling trends were shown), and the increased computation overhead, which was not fully discussed. Reviewers also raised concerns about the hit selection mechanism in cases where no correct answers are generated, as well as the rationale for using incorrect responses as guidance hits for easier questions. Additionally, the paper did not include experiments using the Qwen2.5 Math-based model or comparisons to prior work on 7B models (e.g., PRIME-EURUS, ACEMATH, rStar).

At the end of the discussion, the authors provided access to the source code in an anonymized repository https://anonymous.4open.science/r/MARGE-ACE8 for reproducibility checks and stated that trained models and checkpoints would be open-sourced soon.

 Following a thorough rebuttal, most concerns were adequately addressed, and the reviewers agreed that the paper meets the acceptance threshold. Thus, acceptance is recommended, provided the proposed improvements are incorporated.